# Time-resolved proximity proteomics uncovers a membrane tension-sensitive caveolin-1 interactome at the rear of migrating cells

Eleanor Martin[1,2†], Rossana Girardello[3†], Gunnar Dittmar[3,4*], Alexander Ludwig[1,2*]

[1]School of Biological Sciences, Nanyang Technological University, Singapore, Singapore; [2]NTU Institute of Structural Biology (NISB), Nanyang Technological University, Singapore, Singapore; [3]Proteomics of Cellular Signaling, Luxembourg Institute of Health, Strassen, Luxembourg; [4]Department of Life Sciences and Medicine, University of Luxembourg, Esch-sur-Alzette, Luxembourg

**Abstract** Caveolae are small membrane pits with fundamental roles in mechanotransduction. Several studies have shown that caveolae flatten out in response to increased membrane tension, thereby acting as a mechanosensitive membrane reservoir that buffers acute mechanical stress. Caveolae have also been implicated in the control of RhoA/ROCK-mediated actomyosin contractility at the rear of migrating cells. However, how membrane tension controls the organisation of caveolae and their role in mechanotransduction remains unclear. To address this, we systematically quantified protein–protein interactions of caveolin-1 in migrating RPE1 cells at steady state and in response to an acute increase in membrane tension using biotin-based proximity labelling and quantitative mass spectrometry. Our data show that caveolae are highly enriched at the rear of migrating RPE1 cells and that membrane tension rapidly and reversibly disrupts the caveolar protein coat. Membrane tension also detaches caveolin-1 from focal adhesion proteins and several mechanosensitive regulators of cortical actin including filamins and cortactin. In addition, we present evidence that ROCK and the RhoGAP ARHGAP29 associate with caveolin-1 in a manner dependent on membrane tension, with ARHGAP29 influencing caveolin-1 Y14 phosphorylation, caveolae rear localisation, and RPE1 cell migration. Taken together, our work uncovers a membrane tension-sensitive coupling between caveolae and the rear-localised F-actin cytoskeleton. This provides a framework for dissecting the molecular mechanisms underlying caveolae-regulated mechanotransduction pathways.

*For correspondence:
Gunnar.Dittmar@lih.lu (GD);
aludwig@ntu.edu.sg (AL)

†These authors contributed equally to this work

Competing interest: The authors declare that no competing interests exist.

## Editor's evaluation

This important study uses powerful time-resolved proximity proteomics, validated with proximity ligation assays, to provide new insights into the mechanical regulation of caveolin-1 complexes that form in migrating cells. Follow-up experiments reveal a reciprocal relationship between mechanosensitive caveolae and RhoGTPase signaling in migrating cells. This work is generally convincing, with the exception that the functional link between ARHGAP29 and caveolae under low membrane tension would benefit from a more detailed characterization in the future

## Introduction

Caveolae are an abundant feature of the plasma membrane of almost all vertebrate cells. They are composed of the scaffolding proteins caveolin-1 (Cav1) and caveolin-2 (Cav2), which are embedded in the inner leaflet of the plasma membrane, and the cavin protein family, which form a loose peripheral

caveolar membrane coat. There are four cavins in humans (cavin1/PTRF, cavin2/SDPR, cavin3/SRBC, and cavin4/MURC). Cav1 and cavin1 are essential for caveolae formation, whereas cavins 2–4 regulate caveolae dynamics and functions in a cell and tissue-specific fashion (*Kovtun et al., 2015*; *Parton et al., 2020*; *Shvets et al., 2014*). A third caveolin protein (Cav3) and cavin4 are expressed exclusively in muscle cells. In addition, the ATPase EHD2 and the F-BAR domain containing protein Pacsin2 are found at caveolae. While caveolins and cavins assemble into an oligomeric membrane coat that shapes the caveolar bulb (*Gambin et al., 2014*; *Kovtun et al., 2015*; *Kovtun et al., 2014*; *Ludwig et al., 2013*; *Ludwig et al., 2016*; *Stoeber et al., 2016*), EHD2 and Pacsin2 are localised to the caveolar neck and link caveolae to actin filaments to stabilise caveolae at the plasma membrane and to restrict their mobility (*Hansen et al., 2011*; *Ludwig et al., 2013*; *Morén et al., 2012*; *Senju et al., 2011*; *Stoeber et al., 2012*).

Caveolae are extraordinarily dynamic in nature and have been implicated in diverse biological processes (*Cheng and Nichols, 2016*; *Parton and Simons, 2007*). Caveolae are abundant in cells that are exposed to tensile forces or shear stress, such as endothelial cells, epithelial cells, and muscle cells, and protect such cells from mechanical stress (*Chai et al., 2013*; *Cheng et al., 2015*; *Dewulf et al., 2019*; *Garcia et al., 2017*; *Lim et al., 2017*; *Lo et al., 2015*; *Lu et al., 2017*). There is evidence that caveolae disassemble (*Sinha et al., 2011*) or flatten out (*Matthaeus et al., 2022*) in response to an increase in membrane tension, and may similarly respond to other stresses such as osmotic stress (*Guo et al., 2015*), oxidative stress (*Mougeolle et al., 2015*), and UV irradiation (*McMahon et al., 2019*). Tension-induced flattening was proposed to be mediated (or accompanied) by the dissociation of the peripheral cavin coat and EHD2 from membrane-embedded caveolin oligomers (*Sinha et al., 2011*; *Yeow et al., 2017*). Cavin complexes and EHD2 released into the cytosol can subsequently translocate into the nucleus to regulate gene transcription (*McMahon et al., 2021*; *McMahon et al., 2019*; *Torrino et al., 2018*). This suggests that caveolae act as a dynamic mechanosensitive membrane reservoir that senses changes in membrane tension and transmits such inputs to downstream signalling (*Del Pozo et al., 2021*; *Echarri and Del Pozo, 2015*; *Golani et al., 2019*; *Nassoy and Lamaze, 2012*; *Parton and del Pozo, 2013*).

Large numbers of caveolae are also found at the rear of migrating cells (*Hetmanski et al., 2019*; *Ludwig et al., 2013*), and several studies indicate an intricate relationship between caveolae-mediated mechanosensing and RhoA-mediated cell rear retraction (*Grande-García and del Pozo, 2008*; *Grande-García et al., 2007*; *Hetmanski et al., 2019*). Caveolae formation at the cell rear is promoted by low membrane tension and is dependent upon RhoA/ROCK1 signalling, the Rho guanidine nucleotide exchange factor (GEF) Ect2, and the RhoA effector protein and serine-threonine kinase PKN2 (*Hetmanski et al., 2019*). Caveolae, in turn, enhance RhoA/ROCK1 signalling, leading to F-actin alignment, actomyosin contractility, and rapid rear retraction. Loss of caveolae, increased membrane tension, or inhibition of RhoA signalling all break this positive feedback loop, inhibiting rear retraction and impeding cell migration (*Hetmanski et al., 2019*; *Hetmanski et al., 2021*). This indicates a central role for caveolae in coupling changes in membrane tension to the control of actomyosin contractility at the rear of migrating cells. However, how membrane tension controls caveolae assembly and disassembly and caveolae-mediated signalling at the cell rear has not been analysed in a quantitative and objective manner.

Here we used live-cell imaging and electron tomography to study the dynamics and ultrastructure of caveolae at the rear of migrating RPE1 cells. We then employed proximity biotinylation with the peroxidase APEX2 and quantitative proteomics to define a Cav1-associated interactome in RPE1 cells at iso-osmotic conditions and in response to an acute increase in membrane tension elicited by hypo-osmotic shock. Our data demonstrate that membrane tension rapidly and reversibly disassembles the caveolae structure and abolishes the linkage between caveolae and the cortical actin cytoskeleton. In addition, we identify a number of potentially novel regulators and effectors of caveolae and present evidence that one of them, the Rho GTPase activating protein (GAP) ARHGAP29, controls caveolae rear localisation and RPE1 cell migration.

## Results

### Caveolae are abundant at and stably associated with the rear of RPE1 cells

We previously demonstrated that caveolae are enriched at the rear of hTERT-RPE1 cells (henceforth referred to as RPE1 cells) (*Ludwig et al., 2013*). Quantification of endogenous Cav1 fluorescence intensity in fixed RPE1 cells (visualised with anti-Cav1 antibodies) revealed an approximately eightfold enrichment at the cell rear compared to the cell front (*Figure 1A*). Such a polarised rear distribution of Cav1 is observed in ~70% of RPE1 cells sparsely grown on fibronectin-coated glass. To study the dynamics of caveolae in migrating RPE1 cells, we generated cell lines stably expressing fluorescently tagged cavin1 or cavin3 fusion proteins. Cavin1-APEX2-EGFP (Cavin1-A2E) was concentrated at the cell rear and colocalised with endogenous Cav1, as expected (*Figure 1B*, *Figure 1—figure supplement 1A and B*). Time-lapse microscopy and quantitative tracking of caveolae localisation in RPE1 cells migrating on a fibronectin-coated glass surface showed that both cavin1 and cavin3 fusion proteins were stably associated with the cell rear for several hours (*Figure 1C and D*, *Figure 1—videos 1–3*, *Figure 1—figure supplement 1*). Moreover, caveolae were associated with the rear of RPE1 cells migrating in a 3D collagen matrix (*Figure 1E*, *Figure 1—video 4*). Persistently migrating RPE1 cells showed relatively uniform protrusion and retraction speeds, resulting in an average translocation rate of ~0.4 µm/min. This is consistent with a previous report (*Vaidžiulytė et al., 2022*) and comparable to cell migration speeds of MEFs and MDCK cells and A2780 cells migrating in durotactic gradients (*Hetmanski et al., 2019*; *Sitarska and Diz-Muñoz, 2020*). When cells re-oriented their front-rear axis, this uniform protrusion/retraction cycle was transiently interrupted. The cells ceased to move and waves of new protrusions formed to establish a new leading edge. Interestingly, during this protrusion phase the rear remained relatively static, leading to a slow but persistent increase in cell length (aspect ratio) and cell area. At a certain point, the rear suddenly retracted, causing instantaneous and large cell rear displacements and an apparent accumulation of caveolae at the rear (*Figure 1F–H*, *Figure 1—figure supplement 1D and F*). Complete reversals of the front-rear axis were also frequently observed. In such cases, the polarised distribution of caveolae was transiently lost but was rapidly re-established after a new front-rear axis had been established (*Figure 1—video 7*). Similarly, caveolae rear localisation was rapidly re-established after cell division (*Figure 1—video 8*). Our live-cell imaging data further show that when cells polarise spontaneously, caveolae become aligned along the long cell axis in a filament-like pattern and eventually accumulate at the rear membrane as the rear retracts and the polarity axis is established (*Figure 1—figure supplement 2*, *Figure 1—videos 9 and 10*). Together this indicates that caveolae are stably associated with the RPE1 cell rear and suggests that rear retraction promotes the accumulation of caveolae at the trailing edge.

Next we used RPE1 cells stably expressing a cavin3-miniSOG-mCherry fusion protein to visualise the 3D architecture of caveolae at the cell rear through miniSOG labelling and 3D electron tomography (*Ludwig et al., 2013*; *Shu et al., 2011*). 3D tomography and surface rendering of the tomograms revealed a dense network of surface-connected caveolae and the presence of large interconnected caveolar clusters, most of which appeared to be continuous with the rear plasma membrane (*Figure 2A and B*, *Figure 2—video 1*). We concluded that caveolae are abundant at the rear of RPE1 cells, which hence provide an appropriate model cell type to study the caveolae-associated protein network in migrating cells.

### The caveolin-1 interactome is enriched in cortical actin regulators and is regulated by membrane tension

Having established that caveolae are enriched at the cell rear, we aimed to identify proteins associated with caveolae in migrating cells and ask how such a protein network would respond to an acute increase in membrane tension. Several cytoplasmic effectors of cavins have previously been identified using BioID-mediated proximity proteomics (*McMahon et al., 2021*; *McMahon et al., 2019*; *Mendoza-Topaz et al., 2018*). However, BioID requires long labelling times (16–24 hr) and therefore is not suitable to dissect rapid changes in the caveolae-associated protein network that are likely to occur as cells respond to acute mechanical stress. To address this we employed a proximity biotinylation approach using the APEX2 peroxidase (*Hung et al., 2016*; *Lam et al., 2015*). A stable RPE1 cell line expressing a Cav1-APEX2-EGFP (Cav1-A2E) fusion protein was generated. We have previously

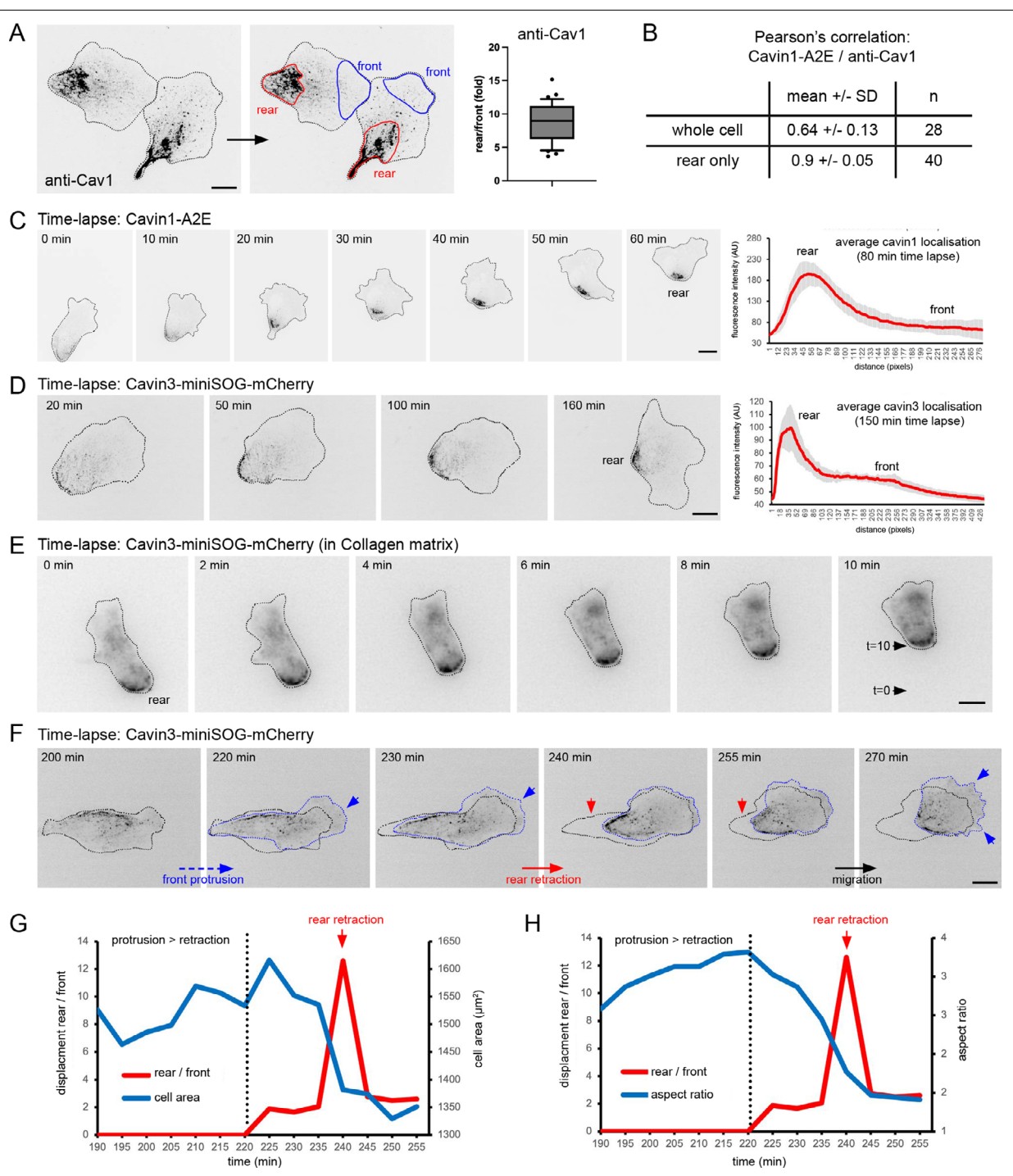

**Figure 1.** Dynamics of caveolae at the rear of migrating RPE1 cells. (**A**) RPE1 cells were fixed and stained with anti-Cav1 antibodies. The rear localisation index (fold enrichment at the rear) was determined by measuring the Cav1 intensity at the cell front and the cell rear (n = 36 cells). Scale bar 10 μm. (**B**) Pearson's correlation analysis of fixed RPE1 cells stably expressing cavin1-A2E stained with anti-Cav1 antibodies. (**C**) Time-lapse images of RPE1 cells stably transfected with Cavin1-A2E migrating on a glass coverslip. The front-rear distribution of Cavin1-A2E in each frame of the movie was measured using line scans. The mean localisation was plotted. Error bars indicate SEM. Scale bar 10 μm. (**D**) Time-lapse images of RPE1 cells stably transfected with Cavin3-miniSOG-mCherry migrating on a glass coverslip. The front-rear distribution of Cavin3 in each frame of the movie was measured using line scans. The mean localisation was plotted. Error bars indicate SEM. Scale bar 10μm. (**E**) Time-lapse images of RPE1 cells stably transfected with cavin3-miniSOG-mCherry migrating on in a collagen matrix. Scale bar 5 μm. (**F**) Time-lapse images of RPE1 cells stably transfected with cavin3-miniSOG-mCherry showing a sudden rear retraction. Scale bar 10μm. (**G**) Temporal relationship between front protrusion, rear retraction, and cell area of the cell shown in (**F**). (**G**) Temporal relationship between front protrusion, rear retraction, and aspect ratio (cell shape) of the cell shown in (**F**).

*Figure 1 continued on next page*

*Figure 1 continued*

The online version of this article includes the following video and figure supplement(s) for figure 1:

**Figure supplement 1.** Characterisation of caveolae rear localisation in RPE1 cells.

**Figure supplement 2.** Caveolae become transiently aligned along the front/rear axis in migrating RPE1 cells.

**Figure 1—video 1.** Time-lapse microscopy of a persistently migrating RPE1 cell stably expressing cavin1-APEX2-EGFP (related to *Figure 1C*).
https://elifesciences.org/articles/85601/figures#fig1video1

**Figure 1—video 2.** Time-lapse microscopy of a persistently migrating RPE1 cell stably expressing cavin3-miniSOG-mCherry (related to *Figure 1D*).
https://elifesciences.org/articles/85601/figures#fig1video2

**Figure 1—video 3.** Time-lapse microscopy of a persistently migrating RPE1 cell stably expressing cavin3-miniSOG-mCherry (related to *Figure 1— figure supplement 1C*).
https://elifesciences.org/articles/85601/figures#fig1video3

**Figure 1—video 4.** Time-lapse microscopy of an RPE1 cell stably expressing cavin3-miniSOG-mCherry persistently migrating in a 3D collagen matrix (related to *Figure 1E*).
https://elifesciences.org/articles/85601/figures#fig1video4

**Figure 1—video 5.** Time-lapse microscopy of an RPE1 cell stably expressing cavin3-miniSOG-mCherry.
https://elifesciences.org/articles/85601/figures#fig1video5

**Figure 1—video 6.** Time-lapse microscopy of an RPE1 cell stably expressing cavin3-miniSOG-mCherry.
https://elifesciences.org/articles/85601/figures#fig1video6

**Figure 1—video 7.** Time-lapse microscopy of an RPE1 cell stably expressing cavin3-miniSOG-mCherry undergoing a complete reversal of front-rear polarity.
https://elifesciences.org/articles/85601/figures#fig1video7

**Figure 1—video 8.** Time-lapse microscopy of an RPE1 cell stably expressing cavin3-miniSOG-mCherry undergoing cell division.
https://elifesciences.org/articles/85601/figures#fig1video8

**Figure 1—video 9.** Time-lapse microscopy of an RPE1 cell stably expressing cavin3-miniSOG-mCherry undergoing front-rear polarisation.
https://elifesciences.org/articles/85601/figures#fig1video9

**Figure 1—video 10.** Time-lapse microscopy of an RPE1 cell stably expressing cavin3-miniSOG-mCherry.
https://elifesciences.org/articles/85601/figures#fig1video10

shown that expression of this construct rescues caveolae formation in mouse embryonic fibroblasts from *CAV1* -/- mice, indicating that this fusion protein is functional (*Ludwig et al., 2017*; *Ludwig et al., 2016*). Cav1-A2E in RPE1 cells was expressed at levels comparable to or lower than that of endogenous Cav1 (*Figure 3A*), colocalised with PTRF/cavin1 at the cell rear (*Figure 3B*, *Figure 3— figure supplement 1A*), and was stably associated with the rear in migrating cells (*Figure 3—figure supplement 1B*, *Figure 3—video 1*). Moreover, transmission electron microscopy showed that the Cav1-A2E fusion protein was efficiently incorporated into caveolae (*Figure 3C and D*).

Next we ascertained that proximity biotinylation in the Cav1-A2E cell line is specific and spatially restricted. Streptavidin-HRP blotting of Cav1-A2E RPE1 cell lysates showed that the biotinylation reaction was dependent upon the presence of biotin phenol and a brief (1 min) exposure to hydrogen peroxide (*Figure 3E*). In addition, fluorescent streptavidin labelling showed that the biotinylation reaction was restricted to the cell rear and colocalised with Cav1-A2E (*Figure 3F*). We also generated a control RPE1 cell line expressing APEX2-EGFP fused to a nuclear export signal (NES-A2E). In this cell line biotinylated proteins were diffusely localised in the cytoplasm, as shown previously (*Figure 3— figure supplement 1C–F*; *Tan et al., 2020a*).

To generate Cav1 proximity proteomes at resting conditions and upon an acute increase in membrane tension, we used a hypo-osmotic shock assay (*Sinha et al., 2011*; *Figure 4A*). Cav1-A2E-expressing cells were either grown in isotonic medium and left untreated (non-treated; NT), exposed to a brief 5 min hypo-osmotic shock (HYPO), or exposed to hypo-osmotic shock and allowed to recover in iso-osmotic medium for 30 min (REC). NES-A2E cells grown in iso-osmotic medium (NT) were used as a reference (or 'spatial ruler') to subtract abundant cytoplasmic proteins and non-specific bystanders. After ratiometric filtering over the NES sample, we identified a total of 348 unique proteins that were significantly enriched in the three Cav1 proximity proteomes (see *Supplementary file 1* for the mass spectrometry data). In total, 70 proteins were exclusive to the HYPO sample, 15 proteins were enriched specifically in the HYPO and REC samples, and 26 proteins were enriched specifically

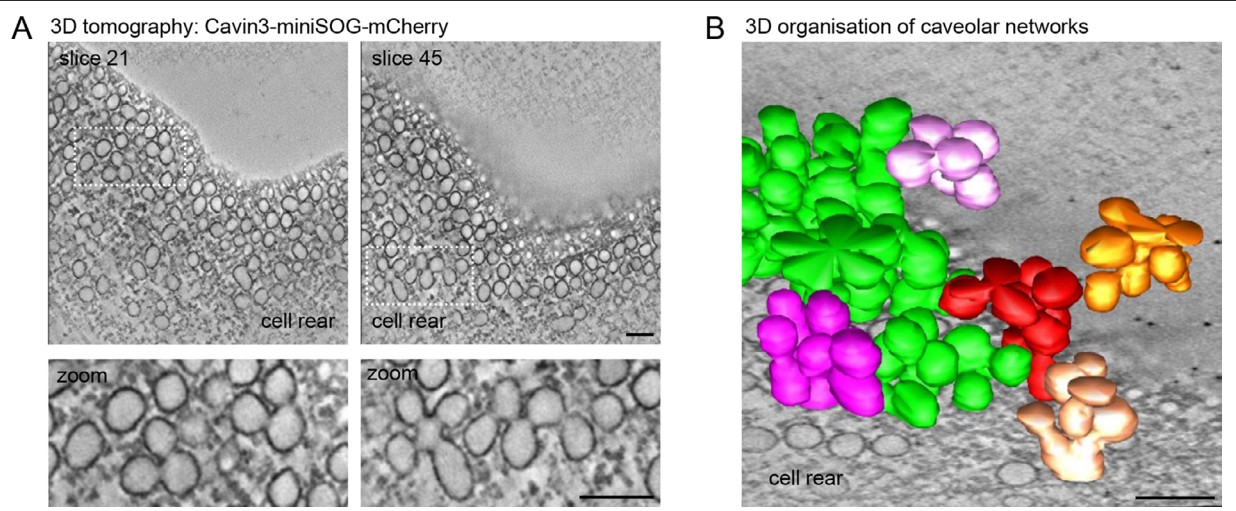

**Figure 2.** Electron tomography of caveolae at the RPE1 cell rear. (**A**) Electron tomography of caveolae and caveolar networks at the rear of RPE1 cells stably transfected with Cavin3-miniSOG-mCherry. Two slices of the tomogram are shown. Note that individual caveolae are interconnected. (**B**) 3D surface model of the tomogram shown in (**A**), showing interconnected clusters, or rosettes, of caveolae. Scale bar in (**A**) and (**B**) is 100 nm.

The online version of this article includes the following video for figure 2:

**Figure 2—video 1.** Electron tomogram and surface rendering of caveolar clusters and networks at the rear of RPE1 cells stably expressing cavin3-miniSOG-mCherry.

https://elifesciences.org/articles/85601/figures#fig2video1

in the NT and REC samples (*Figure 4B*, *Figure 4—figure supplement 1*). Importantly, Cav1-A2E and all major caveolar proteins (Cav1, Cav2, PTRF/cavin1, and EHD2; note that cavin2 and cavin3 are not expressed in RPE1 cells) were specifically enriched in the iso-osmotic control (NT) and recovered (REC) samples. On the contrary, under hypo-osmotic (HYPO) conditions, Cav1, Cav2, and EHD2 were no longer significantly enriched with Cav1-A2E (*Figure 4C*), suggesting that high membrane tension had caused the dissociation of these proteins from Cav1-A2E. Indeed, Cav1-A2E, PTRF/cavin1, and EHD2 were all specifically enriched in the NT and REC samples when directly compared to the HYPO sample (*Figure 4D*). By contrast, no significant differences in the abundance of caveolar core components were observed between NT and REC samples, suggesting that caveolae formation is fully restored within 30 min of recovery from a hypo-osmotic shock. We concluded that proteins identified specifically in the NT and REC samples are associated with Cav1 in a membrane-tension-dependent manner and therefore constitute a reliable Cav1 proximity proteome under iso-osmotic conditions.

Quantification of the MS data and GO annotation analyses revealed that the Cav1 proximity proteome was enriched in components of the cortical actin cytoskeleton and integrin adhesions, including filamins (FLNA, FLNB, FLNC), utrophin (UTRN), tensin-1 (TNS1), cortactin (CTTN), and paxillin (PXN) (*Figure 4E and G*, *Figure 4—figure supplement 1*). Of note, all caveolar proteins showed negative Z-scores (i.e., were under-represented) under hypo-osmotic conditions (*Figure 4F*). Some proteins such as Pacsin2, utrophin, and tensin-1 were entirely absent from the Cav1 proximity proteome obtained in hypo-osmotic medium (*Figure 4E*; indicated as NA). A comprehensive database and literature review further identified several direct and indirect interactions between caveolar proteins, filamins and focal adhesion components (*Figure 3H*). These data support previous work demonstrating a role for caveolae in focal adhesion turnover and integrin-mediated mechanotransduction (*Joshi et al., 2008*; *Lolo et al., 2022*; *Meng et al., 2017*; *Nethe and Hordijk, 2011*). Moreover, the F-actin crosslinking protein filamin-A and utrophin are well-known mechanotransducers (*Rajaganapathy et al., 2019*; *Stossel et al., 2001*) and have been shown to regulate caveolae trafficking and dynamics, possibly by linking caveolae to the actin cytoskeleton via direct or indirect interactions with Cav1 or PTRF/cavin1 (*Kanai et al., 2014*; *Muriel et al., 2011*; *Palma-Flores et al., 2014*; *Stahlhut and van Deurs, 2000*; *Sverdlov et al., 2009*). Filamins and utrophin, as well as caldesmon-1 (CALD1) and calponin-2 (CNN2), have also all been implicated in the control of cell migration (*Baldassarre et al., 2009*; *Hossain et al., 2016*; *Li et al., 2013*; *Liu and Jin, 2016*; *Mayanagi and Sobue, 2011*; *Sobue*

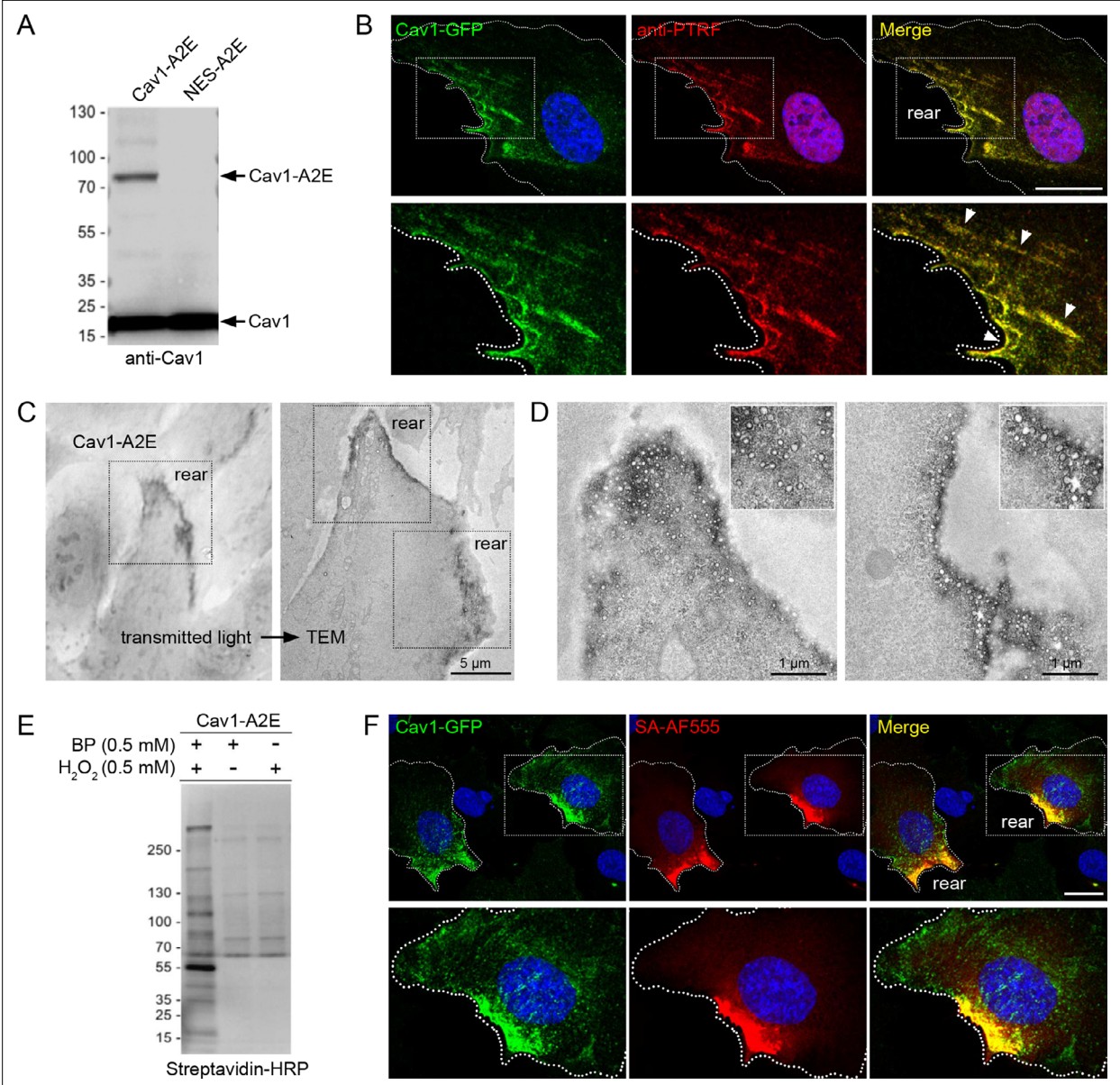

**Figure 3.** Proximity labelling of caveolae in RPE1 cells stably expressing Cav1-APEX2-EGFP. (**A**) Western blot of stable RPE1 cell lines expressing either Cav1-A2E or NES-A2E fusion proteins. The membrane was probed with anti-Cav1 antibodies. (**B**) Confocal fluorescence microscopy of Cav1-A2E RPE1 cells stained with anti-PTRF/cavin1 antibodies. Boxed regions are magnified in the bottom panel. Nuclei were counterstained with DAPI. Scale bar 10 μm. (**C**) Transmitted light image of Cav1-A2E RPE1 cells after APEX2 labelling and plastic embedding (left) and correlative TEM image of the same cell (right). (**D**) Representative TEM micrographs of serial TEM sections of the cell shown in (**C**) (boxed regions). Note the specific labelling and abundance of caveolae at the cell rear. (**E**) Cav1-A2E-expressing cells were incubated in the absence or presence of biotin phenol and/or hydrogen peroxide. Cell lysates were separated by SDS-PAGE, blotted onto PVDF membrane, and probed with streptavidin-HRP. (**F**) Confocal microscopy of Cav1-A2E RPE1 cells after proximity labelling with APEX2. Cells were fixed and stained with fluorescent streptavidin (Alexa Fluor 555) to visualise biotinylated proteins. Nuclei were counterstained with DAPI. Scale bar 10 μm.

The online version of this article includes the following video, source data, and figure supplement(s) for figure 3:

**Source data 1.** Original western blots.

**Figure supplement 1.** Characterisation of the Cav1-A2E RPE1 cell lines.

**Figure supplement 1—source data 1.** Original western blots.

**Figure 3—video 1.** Time-lapse microscopy of a persistently migrating RPE1 cell stably expressing Cav1-APEX2-EGFP (related to *Figure 3—figure supplement 1B*).

https://elifesciences.org/articles/85601/figures#fig3video1

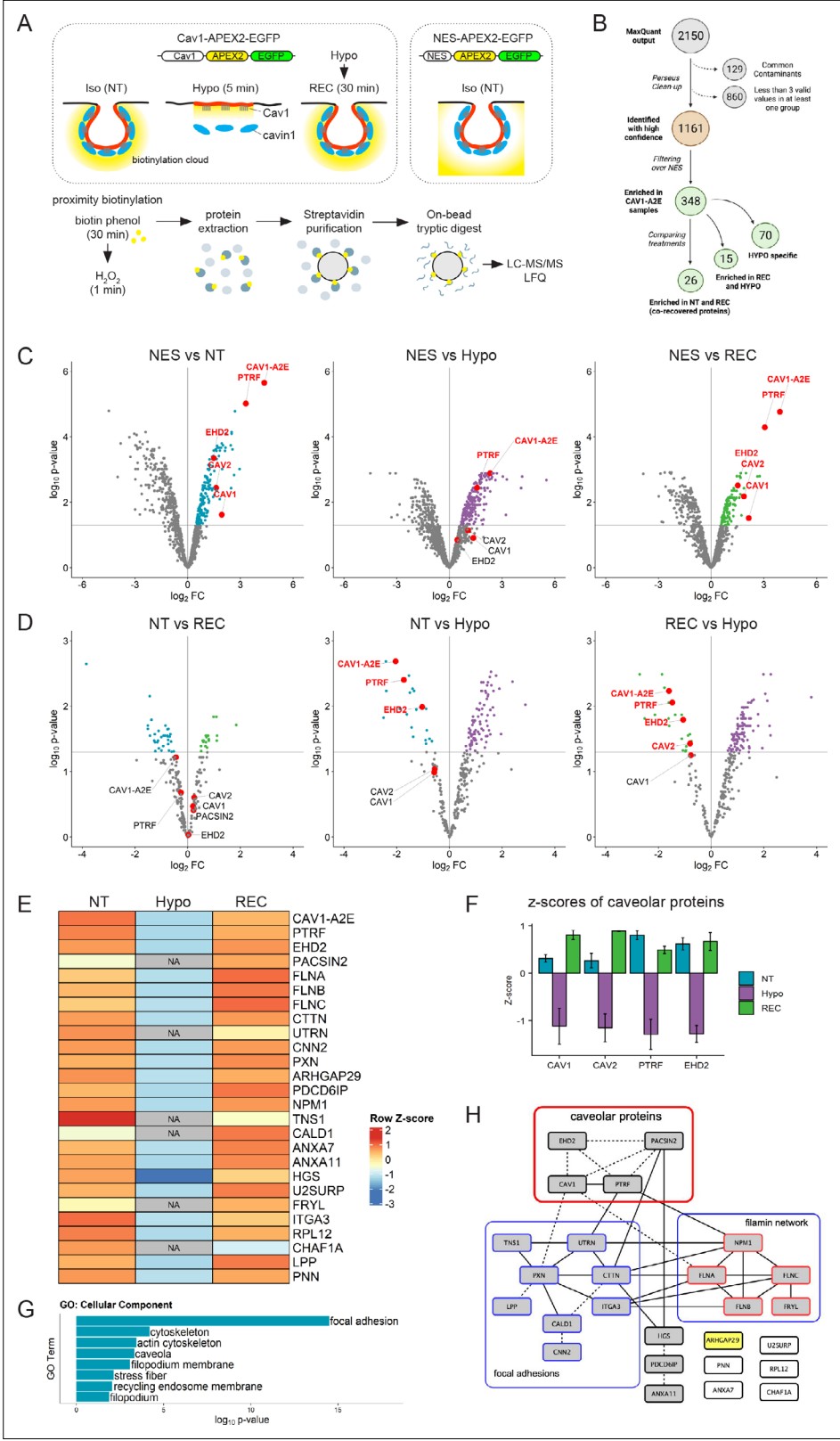

**Figure 4.** The caveolin-1 interactome is enriched in cortical actin regulators and focal adhesion proteins and is highly sensitive to membrane tension. (**A**) Experimental set-up used for APEX2-mediated proximity biotinylation and LC-MS/MS. After proximity labelling, cells were lysed and biotinylated proteins were captured using magnetic streptavidin beads. Purified proteins were on bead-digested with trypsin and the recovered peptides analysed

*Figure 4 continued on next page*

*Figure 4 continued*

by LC-MS/MS. Three individual replicates were analysed. (**B**) Diagram summarising the LC-MS/MS analysis and the total number of proteins significantly enriched in each treatment compared to the NES control. Only proteins identified in all three replicates in at least one of the three samples were considered for further analysis. Label-free quantification (LFQ) was used to define specific Cav1 interactomes and to determine quantitative changes upon hypo-osmotic shock and recovery. (**C**) Volcano plots showing proteins significantly enriched in NT, HYPO, and REC samples compared to the NES control sample. Only proteins with a p-value <0.05 and a logFC > 0 were considered for further analysis. (**D**) Volcano plots comparing the NT and REC, NT and HYPO, and REC and HYPO samples. Cav1-A2E fusion protein and the core caveolar proteins are highlighted. (**E**) Heatmap of z-scored LFQ quantified protein groups differentially enriched across the treatments. NA indicates not identified. See *Figure 3—figure supplement 1* for a complete analysis. (**F**) Z-scores of caveolar proteins under NT, HYPO and REC conditions. (**G**) Gene Ontology (GO) Cellular Component analysis for proteins enriched under NT and REC conditions (see **E**). (**H**) PPI network of the Cav1 proximity proteome. Only proteins significantly enriched in both the NT and REC samples were considered (i.e., 26 co-recovered proteins). Solid lines indicate interactions retrieved from databases (BioGRID, Reactome, BioCarta), dotted lines indicate interactions retrieved from a hand-curated literature search (see *Supplementary file 2*).

The online version of this article includes the following figure supplement(s) for figure 4:

**Figure supplement 1.** Quantitative proximity proteomics of Cav1-APEX2-EGFP in response to hypo-osmotic shock.

*and Sellers, 1991*; *Ulmer et al., 2013*). We concluded that time-resolved proximity labelling with APEX2 uncovered a dynamic Cav1 interactome enriched in mechanosensitive cortical actin regulators, which is highly sensitive to acute changes in membrane tension.

To validate our proteomics data, we quantified the spatial proximity between Cav1 and some of the identified proteins using proximity ligation assays (PLA) (*Figure 5A–F*, *Figure 5—figure supplement 1*). In agreement with our proteomics data, under iso-osmotic conditions we observed robust PLA signals for Cav1-A2E and PTRF/cavin1, Cav1-A2E and filamin-A, and Cav1-A2E and cortactin (*Figure 5A–D*). Moreover, robust signals were observed for endogenous Cav1 and an APEX2-EGFP fusion protein of EHD2 (EHD2-A2E) (*Figure 5E and F*). As expected, 70–80% of the PLA signals for Cav1-A2E and PTRF/cavin1 and anti-Cav1 and EHD2-A2E localised to the cell rear, whilst rear-localised PLA signals for Cav1-A2E and filamin-A and Cav1-A2E and cortactin were significantly lower (~40%) (*Figure 5G*). This is expected as filamin-A and cortactin were present but not enriched at the cell rear (*Figure 5—figure supplement 1*). Importantly, hypo-osmotic shock drastically reduced the PLA signals in all of the above cases, whereas cells that had recovered from the hypo-osmotic shock showed PLA signals indistinguishable to that of control cells. Taken together our data indicate that a brief hypo-osmotic shock causes an instantaneous but reversible disassembly of caveolae, as shown previously (*Sinha et al., 2011*; *Torrino et al., 2018*; *Yeow et al., 2017*). The data further imply that membrane tension regulates the linkage between caveolae and the cortical actin cytoskeleton and that caveolae assembly/disassembly is spatio-temporally coordinated with the reorganisation of the rear-localised actin cortex.

## ROCK activity is required for caveolae reassembly upon hypo-osmotic shock

Interestingly, several proteins were specifically enriched with Cav1 during the recovery phase (i.e., in HYPO and REC samples) (*Figure 4—figure supplement 1*). Among these proteins were Rho-associated protein kinase 1 (ROCK1) and the heat shock protein HSPB1. Rho/ROCK1 signalling was previously shown to be required for caveolae localisation to the rear of migrating cells (*Hetmanski et al., 2019*) and HSPB1 is a stress-regulated actin binding protein implicated in mechanotransduction and cell migration (*Clarke and Mearow, 2013*; *Collier et al., 2019*; *Hoffman et al., 2022*). PLAs confirmed that the association of both HSPB1 and ROCK1 with Cav1 is significantly enhanced in cells exposed to hypo-osmotic shock and in cells that had recovered from the shock (*Figure 5—figure supplement 2*). In addition, pathway enrichment analysis suggested a functional link between Cav1, ROCK1, and HSPB1 (*Figure 5—figure supplement 2D*). To test whether ROCK1 activity, and hence actomyosin contractility, is required for caveolae reformation following hypo-osmotic shock, we performed PLAs in the absence or presence of the ROCK inhibitor Y-27632. ROCK inhibition blocked caveolae rear

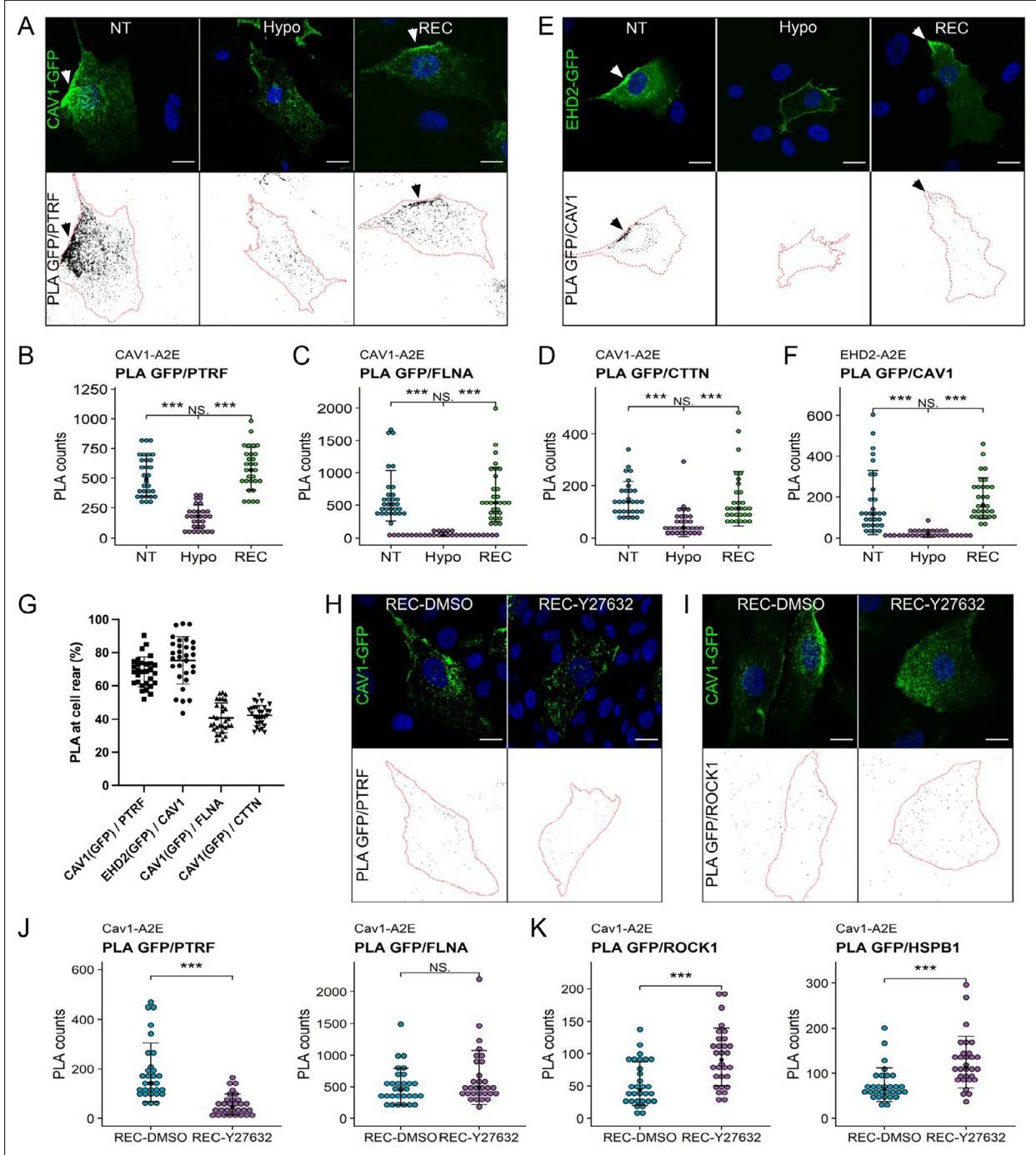

**Figure 5.** Proximity ligation assays (PLA) reveal dynamic interactions between caveolae and the cortical actin cytoskeleton during changes in membrane tension. (**A–D**) PLA in the Cav1-A2E RPE1 cell line using anti-GFP and anti-PTRF/cavin1 (**A, B**), anti-FLNA (**C**), or anti-CTTN (**D**) antibodies as indicated. (**E, F**) PLA in RPE1 cells transfected with EHD2-A2E using anti-GFP and anti-Cav1 antibodies. (**G**) Quantification of rear-localised PLA signals based on data shown in panels (**A–F**). (**H–K**) Effect of ROCK inhibition on PLA in the Cav1-A2E RPE1 cell line using the indicated combinations of antibodies. Cells were incubated for 5 min in hypo-osmotic medium containing 10 μM Y-27632 or DMSO (control) and then transferred to isotonic medium for 30 min in the absence (DMSO) or presence of Y-27632. The graphs in (**B, C, D, F, J, K**) show the quantifications of the PLA signals per cell. 30 cells from three independent experiments were quantified for each treatment. The number of PLA dots per cell is presented with the mean values ± SD. The bottom panels in (**A, E, H, I**) show the PLA signal masks used for counting. Statistical significance was calculated using a Wilcoxon test. ns = p>0.05, ***p≤0.001. All scale bars 20 μm.

The online version of this article includes the following figure supplement(s) for figure 5:

**Figure supplement 1.** Proximity ligation assays (PLA) in Cav1-A2E RPE1 cells using CTTN and FLNA antibodies.

*Figure 5 continued on next page*

Figure 5 continued

**Figure supplement 2.** Proximity ligation assays (PLA) in Cav1-A2E RPE1 cells using ROCK1 and HSPB1 antibodies.

localisation (*Figure 5H and I*) and significantly reduced the PLA signal between Cav1-A2E and PTRF/cavin1, but did not affect the Cav1-A2E/filamin-A interaction (*Figure 5J*). Interestingly, PLA signals between Cav1-A2E and ROCK1 as well as between Cav1-A2E and HSPB1 were significantly increased upon ROCK inhibition (*Figure 5K*). This corroborates our proximity proteomics data and suggests that (a) ROCK1 and HSPB1 are recruited to caveolae (or the cell rear) at high membrane tension, (b) that ROCK activity is required for caveolae reassembly after hypo-osmotic shock, and (c) that ROCK activity weakens the protein's association with caveolae (or the cell rear).

## ARHGAP29 controls caveolin-1 phosphorylation, caveolae rear localisation, and RPE1 cell migration

A number of proteins identified in the Cav1 interactome had previously not been linked to caveolae (*Figure 4E and H*). One of these proteins, the RhoGAP ARHGAP29, has been shown to regulate mechanotransduction in a YAP-dependent manner and to control cancer cell migration and invasion by suppressing actin polymerisation through a Rho/LIMK/cofilin pathway (*Meng et al., 2018*; *Qiao et al., 2017*). YAP also regulates caveolar gene transcription and caveolae formation (*Rausch et al., 2019*), and there is evidence that caveolae, in turn, regulate YAP (*Moreno-Vicente et al., 2019*; *Rausch et al., 2019*). This might suggest a functional link between caveolae, ARHGAP29-mediated Rho signalling, and the HIPPO/YAP pathway. To test this, we downregulated ARHGAP29 or Cav1 in RPE1 cells using specific siRNAs (*Figure 6A and B*). Interestingly, ARHGAP29, YAP, and pS127 YAP levels (a common readout for HIPPO pathway activity) were significantly increased in Cav1 siRNA cells. In addition, the levels of phosphorylated myosin light chain (pMLC, a common readout for myosin-II activation) were markedly decreased in Cav1 siRNA cells. By contrast, ARHGAP29 siRNA transfected cells showed a significant increase in Cav1 Y14 phosphorylation, but pMLC levels were not significantly different to control cells. To test whether the increase in ARHGAP29 protein observed in Cav1 siRNA cells was due to elevated YAP-mediated gene transcription, we performed qPCR analyses. YAP and YAP target gene expression [CYR61 (CCN1), CTGF (CCN2), ANKRD1] as well as ARHGAP29 mRNA levels were unchanged in Cav1 knockdown cells (*Figure 6—figure supplement 1*). Taken together these data suggest a functional link between caveolae and ARHGAP29, which appears to be independent of YAP signalling.

Next we analysed ARHGAP29 siRNA-transfected cells using fixed and live-cell microscopy. The localisation of Cav1 and PTRF/cavin1 was not notably altered in ARHGAP29 siRNA-transfected cells (data not shown). In addition, cells with reduced ARHGAP29 expression showed normal cell migration behaviour compared to control siRNA transfected cells (*Figure 6—figure supplement 2A–D*). The lack of a migration phenotype may be explained by residual ARHGAP29 activity (i.e., incomplete knockdown) or functional redundancy (i.e., compensation by other related RhoGAPs). We therefore analysed the subcellular distribution of Cav1 and PTRF/cavin1 in cells overexpressing moderate levels of ARHGAP29. Only cells with an intact actin cytoskeleton and normal cell shape were considered for the analysis. Transfected A2E-ARHGAP29 (*Figure 6—figure supplement 2E and F*) was localised primarily in the cell cytoplasm, with occasional enrichment in lamellipodia-like protrusions. Interestingly, protrusions enriched in ARHGAP29 exhibited a clear reduction in Cav1 staining intensity. When normalised against total cellular protein levels, such protrusions showed an approximately threefold enrichment of ARHGAP29 over Cav1 (*Figure 6—figure supplement 2E*). In addition, in cells overexpressing ARHGAP29 Cav1 was concentrated in the cell centre rather than being polarised to the rear as observed in cells transfected with the NES-A2E control plasmid (*Figure 6C*). Quantification of this phenotype revealed that overexpression of A2E-ARHGAP29 indeed perturbed the polarised rear distribution of Cav1 (*Figure 6D*). The localisation of PTRF/cavin1 was similarly affected (*Figure 6—figure supplement 2F and G*). These data suggest that ARHGAP29 regulates the polarised distribution of caveolae to the cell rear.

We then investigated the effect of ARHGAP29 overexpression on RPE1 cell migration using time-lapse microscopy (*Figure 6E*). Control-transfected RPE1 cells showed stable front-rear polarity and migrated randomly but persistently, similar to non-transfected cells (*Figure 6E and F*, *Figure 6—video 1*). By contrast, cells overexpressing A2E-ARHGAP29 established multiple random protrusions and

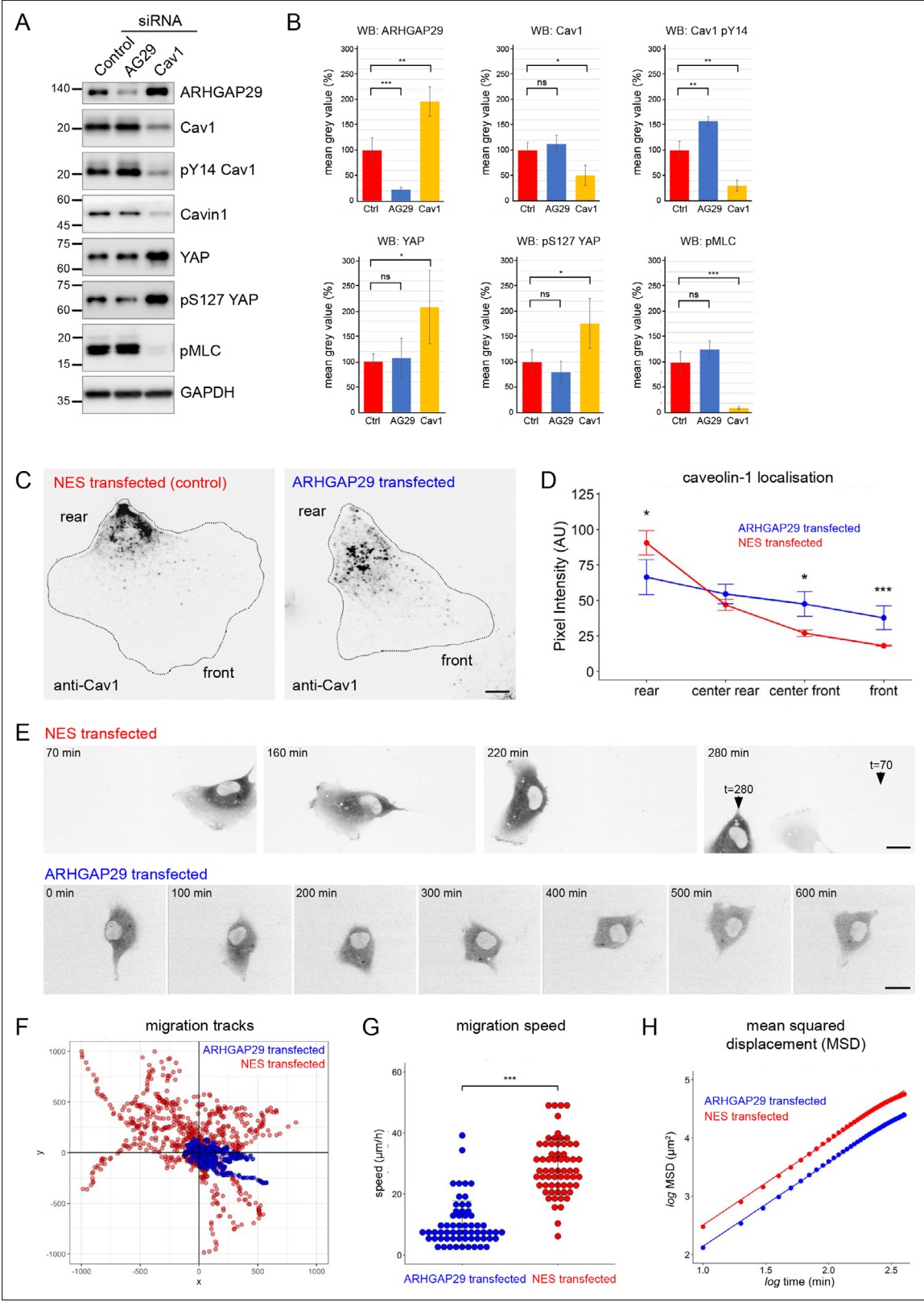

**Figure 6.** Evidence for a functional connection between caveolae and ARHGAP29. (**A**) RPE1 cells were transfected with siRNAs against ARHGAP29, caveolin-1, or non-targeting siRNAs and analysed by western blotting 48 hr post-transfection. (**B**) Quantification of western blot analysis shown in (**A**). Ratios normalised to the GAPDH loading control are displayed relative to the intensity of the control siRNA transfection for each protein indicated. Data represent the mean ± SD of 3–4 independent experiments. Statistical significance was calculated using an unpaired *t*-test. ns = p>0.05, *p≤0.05,

*Figure 6 continued on next page*

*Figure 6 continued*

**p≤0.01, ***p≤0.001. (**C**) RPE1 cells were transfected with A2E-ARHGAP29 or NES-A2E, fixed, and stained with anti-Cav1 antibodies. Scale bar 5 µm. (**D**) Quantification of Cav1 rear localisation based on data shown in (**C**). Error bars indicate mean ± SEM *p≤0.05, ***p≤0.001, Wilcoxon test (n = 18 cells per condition). (**E**) Still images of RPE1 cells transfected with NES-A2E (top) or A2E-ARHGAP29 (bottom) imaged live by spinning disk confocal microscopy. Scale bars 10 µm. (**F–H**) Migration tracks (**F**), migration speed (**G**), and mean squared displacement (**H**) of RPE1 cells transfected with A2E-ARHGAP29 or NES-A2E. Quantification was performed on three independent experiments and a total of ~60 cells per sample. Statistical significance in (**G**) was calculated using an unpaired *t*-test; ***p≤0.001.

The online version of this article includes the following video, source data, and figure supplement(s) for figure 6:

**Source data 1.** Original western blots shown in *Figure 6A* used for the quantification of data shown in *Figure 6B*.

**Source data 2.** Original pMLC western blots shown in *Figure 6A* used for the quantification of pMLC levels shown in *Figure 6B*.

**Figure supplement 1.** Knockdown of Cav1 in RPE1 cells does not affect the transcription of YAP or YAP target genes.

**Figure supplement 2.** Overexpression and siRNA-mediated knockdown of ARHGAP29.

**Figure 6—video 1.** Time-lapse microscopy of RPE1 cells transfected with an NES-APEX2-EGFP plasmid.
https://elifesciences.org/articles/85601/figures#fig6video1

**Figure 6—video 2.** Time-lapse microscopy of RPE1 cells transfected with an APEX2-EGFP-ARHGAP29 plasmid.
https://elifesciences.org/articles/85601/figures#fig6video2

**Figure 6—video 3.** Time-lapse microscopy of RPE1 cells transfected with an APEX2-EGFP-ARHGAP29 plasmid.
https://elifesciences.org/articles/85601/figures#fig6video3

failed to specify a stable front-rear axis, resulting either in a complete loss of cell motility (*Figure 6E and F*, *Figure 6—video 2*) or in short migration tracks and frequent reversals in direction (*Figure 6—video 3*). Consequently, ARHGAP29-overexpressing cells showed markedly reduced migration velocities and mean squared displacements compared to control-transfected cells (*Figure 6G and H*). Taken together the data indicate that ARHGAP29 negatively regulates caveolae rear localisation and RPE1 cell migration.

## Discussion

### Spatially restricted proteomics reveals a membrane-tension-sensitive Cav1 interactome

Using APEX2-mediated proximity proteomics and PLAs, we have identified a distinct Cav1 interactome that is highly sensitive to membrane tension. The data suggest that caveolae link changes in membrane tension to the control of the rear-localised actin cortex, integrin/focal adhesion turnover, and Rho/ROCK signalling (see below). Interestingly, several known mechanotransducers and cell migration regulators including filamins, cortactin, and utrophin were associated with Cav1, suggesting that caveolae integrate mechanosignalling at the rear of migrating cells. We note that the Cav1 interactome is enriched in proteins that are present at but not necessarily exclusive to the cell rear. Given the abundance of caveolae at the rear, the majority of the proteins identified in the interactome are likely to be associated with the rear-localised pool of caveolae. However, Cav1-APEX2 molecules localised elsewhere in the cell, for example in the leading edge, the Golgi apparatus, or on transport vesicles do contribute to the proteome. Therefore, the Cav1 interactome is not a specific interactome of caveolae at the cell rear but rather represents a composite cell-wide interactome of Cav1 oligomers and fully assembled caveolae.

Our proteomics and PLA data show that cavins, EHD2, as well as Pacsin2 are released from Cav1 scaffolds in cells exposed to a brief hypotonic shock. This underscores existing models suggesting that the caveolar coat is rapidly disassembled at high membrane tension (*Del Pozo et al., 2021*). The data also revealed that the entire caveolar coat structure is re-established within 30 min of recovery from the shock, demonstrating that caveolae assembly and disassembly are remarkably dynamic. It is worth noting that flat Cav1 scaffolds have recently been shown to contain detectable levels of cavins and that EHD2 and Pacsin2 are associated both with flat and curved caveolae (*Matthaeus et al., 2022*). It is possible, therefore, that the tension-induced conformational changes in the caveolar coat are more subtle than current models suggest. Nonetheless, our work clearly shows that membrane tension has a profound effect on caveolae and that such changes are quantifiable. Altogether, we suggest that membrane tension either dissociates the caveolar coat partially or induces a substantial

structural remodelling of many but not all caveolae. One notable caveat of our approach is that the level of tension induced by hypo-osmotic shock likely exceeds that exerted at the rear of the cell. Hence, it will be important to address in future how much membrane tension actually changes at the cell rear as cells migrate, and whether these changes are sufficient to cause caveolae disassembly. We also obtained a list of proteins that became selectively enriched with Cav1 at high membrane tension (*Figure 4—figure supplement 1*). This proximity proteome is composed of proteins with diverse cellular functions ranging from protein translation (ribosomal proteins) to membrane trafficking. Whether any of these proteins are involved in mediating a Cav1-dependent mechanosignalling response, for instance, via the recently proposed caveolae-independent Cav1 scaffolds termed dolines (*Lolo et al., 2023*), remains to be tested.

## Membrane tension and actomyosin contractility regulate caveolae rear localisation

We show using fixed and live-cell imaging that caveolar proteins are enriched at and stably associated with the rear of migrating RPE1 cells. Although we cannot discriminate between flat and invaginated caveolae at the resolution provided by diffraction-limited microscopy, large clusters of morphologically distinct caveolae and interconnected caveolar networks were observed at the RPE1 cell rear using electron tomography. Such a polarised distribution of caveolae has been demonstrated in many other migratory cell types (*Grande-García and del Pozo, 2008*; *Grande-García et al., 2007*; *Hetmanski et al., 2019*; *Joshi et al., 2008*). What is the mechanism driving caveolae formation specifically at the cell rear? Several studies have shown that low membrane tension favours caveolae formation (*Del Pozo et al., 2021*; *Golani et al., 2019*; *Sinha et al., 2011*). This suggests that the formation of caveolae specifically at the rear of migrating cells is the result of a relatively stable front-rear asymmetry in membrane tension. Such membrane tension gradients, with low tension at the rear and higher tension at the front, have indeed been observed in fast-moving fish keratocytes (*Lieber et al., 2015*; *Lieber et al., 2013*). However, whether stable membrane tension gradients are a general feature of migrating cells is not entirely clear, and whether membrane tension rapidly equilibrates across the cell (e.g. from the front to the rear) or can be locally confined has been the subject of much debate (*De Belly et al., 2023*; *Shi et al., 2018*; *Sitarska and Diz-Muñoz, 2020*). Using fluorescence lifetime imaging with the Flipper-TR membrane tension probe, the Roux lab recently reported that stable front-rear membrane tension gradients indeed exist in many migrating cells, including RPE1 cells (*García-Arcos et al., 2024*). Tension gradients were even observed in non-migrating cells and were dependent upon a dynamic F-actin network, cell–substrate adhesions, and membrane–cortex attachments. The study provides strong support that membrane tension gradients are a common feature of adherent cells and that the migration mode determines how such gradients are established.

Hetmanski et al. reported that A2780 cells exhibit a front/rear membrane tension gradient only when cells migrate in an ECM rigidity gradient (*Hetmanski et al., 2019*). Both inhibition of Rho signalling by Y-27632 and expression of a constitutively active ezrin mutant (which enhances membrane–cortex interactions at the cell rear) increased rear membrane tension, prevented the accumulation of cavin1 and Cav1 at the rear, and blocked cell rear retraction. The role of contractility in concentrating caveolae at the rear is also evident in our time-lapse movies, which suggest that caveolae (cavins) become concentrated at the rear when cells retract their trailing edge. Such sudden rear retractions are likely the result of bursts of contractility that pull on the rear membrane, lowering tension at the rear and promoting caveolae formation. In addition, cavins were frequently aligned in 'filamentous stripes' when cells establish front/rear polarity or reorient their front/rear axis, indicating a physical linkage between caveolae and F-actin fibres. Caveolae are indeed attached to and aligned along actin filaments (*Echarri and Del Pozo, 2015*; *Richter et al., 2008*; *Stoeber et al., 2012*) and depolymerisation of the actin cytoskeleton or inhibition of actomyosin contractility affects caveolae distribution and dynamics (*Hetmanski et al., 2019*; *Shi et al., 2021*) and blocks caveolae rear localisation (*Hetmanski et al., 2019*). Altogether our data and a large body of existing literature indicate that caveolae are associated with the F-actin cytoskeleton and that actomyosin-driven contraction and a front/rear membrane tension gradient promote caveolae formation specifically at the rear.

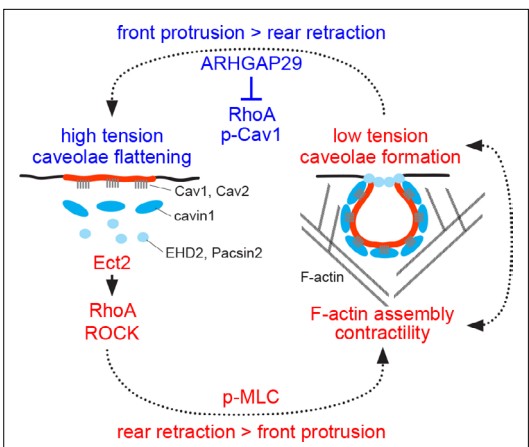

**Figure 7.** Model of caveolae dynamics and function at the rear of migrating cells. Low membrane tension promotes caveolae formation at the cell rear, whilst high membrane tension causes caveolae to flatten out, which is accompanied by the dissociation of cavin1/PTRF, EHD2, and Pacsin2 from membrane-embedded Cav1 scaffolds. The linkage between the cortical F-actin network and Cav1 scaffolds is also lost at high membrane tension. Caveolae or Cav1 scaffolds promote RhoA/ROCK signalling, MLC phosphorylation, and cell rear retraction, possibly by recruiting the RhoGEF Ect2 and ROCK to the cell rear. ARHGAP29 may be recruited to the cell rear at low membrane tension to suppress RhoA signalling, leading to reduced Cav1 Y14 phosphorylation, increased membrane tension, and caveolae flattening.

## A model of caveolae function and dynamics at the cell rear

A putative model of how membrane tension and caveolae regulate actomyosin contractility and cell rear retraction is shown in *Figure 7*. The model assumes that a stable membrane tension gradient exists in RPE1 cells (*García-Arcos et al., 2024*), promoting caveolae formation at the rear. Considering evidence that protrusion at the front is coupled to retraction at the rear and that membrane tension can propagate across the cell (*De Belly et al., 2023*; *Houk et al., 2012*; *Mueller et al., 2017*; *Sitarska and Diz-Muñoz, 2020*; *Tsai et al., 2019*), it can be inferred that at short time scales absolute membrane tension values fluctuate at both the rear and the front. This would suggest that caveolae assembly and disassembly at the rear is coordinated with or driven by the protrusion/retraction cycle: When rear tension drops (e.g. in response to a sudden rear retraction), the formation of caveolae and caveolar networks is favoured. When tension at the rear builds up (e.g. when front protrusion is dominant over rear retraction), caveolae flatten out. Since caveolae and caveolar networks substantially increase the effective membrane surface (*Golani et al., 2019*) and also provide a unique topology and curvature, their dynamic and reversible nature might not only be critical for buffering membrane tension, but also play an important role in regulating the recruitment of signalling proteins to the rear.

Previous studies have demonstrated a critical role for Rho/ROCK signalling in caveolae formation and cell rear retraction (*Hetmanski et al., 2019*; *Hetmanski et al., 2021*). Knockdown of RhoA, ROCK1, or the rear-localised RhoGEF Ect2 in A2780 cells blocked caveolae rear localisation and impeded cell migration in durotactic gradients. Caveolae, in turn, were required for the recruitment of Ect2 to the rear, for RhoA activation, and for cell rear retraction, suggesting that caveolae and the actomyosin system control cell rear retraction via a positive feedback loop (*Hetmanski et al., 2019*). Our data suggest that ROCK1 is recruited to Cav1 at high membrane tension (i.e. under conditions that promote caveolae flattening) and that ROCK activity is required for caveolae reassembly after hypo-osmotic shock. In addition, we show that the loss of Cav1 leads to a drastic reduction in MLC phosphorylation, indicating that Cav1 is critical for myosin-II activation. Altogether this suggests that Cav1 scaffolds recruit Ect2 and ROCK1 to the cell rear to promote caveolae formation and actomyosin contractility. As discussed below, the function of ARHGAP29 might be to interrupt this feedback loop by suppressing RhoA signalling at the rear.

## A putative functional role of ARHGAP29 at the cell rear

We provide evidence that ARHGAP29 is associated with Cav1 in a membrane tension-dependent manner and that overexpression of ARHGAP29 suppresses caveolae rear localisation and RPE1 cell migration. Since caveolae are highly enriched at the cell rear, we expected that ARHGAP29 would (at least partially) be associated with the rear. However, both in fixed and live RPE1 cells ARHGAP29 appeared mostly cytoplasmic, with no obvious rear bias. This was somewhat unexpected as ARHGAP29 is localised to the plasma membrane in several cell types (*Post et al., 2015*; *Rogg et al., 2023*; *Tan et al., 2020b*; *Tan and Zaidel-Bar, 2015*). Closer inspection revealed that a fraction of ARHGAP29 was present in lamellipodia-like protrusions, which interestingly were largely devoid of Cav1. This inverse

spatial correlation suggests that ARHGAP29 can locally suppress caveolae assembly or prevent caveolae recruitment into such areas. This effect is likely due to the local inhibition of RhoA and can explain the caveolae rear polarity and cell migration defects we observe in cells overexpressing ARHGAP29. The apparent absence of ARHGAP29 from the cell rear at steady state then prompts the question why ARHGAP29 was identified in the Cav1 proteome with such specificity and reproducibility in the first place. This can be explained by the way the APEX2 enzyme works. Proximity biotinylation with APEX2 is extremely sensitive and is restricted to a labelling radius of ~20 nm (*Hung et al., 2016*). The labelling reaction is conducted on live and intact cells at room temperature (RT) for 1 min. Although 1 min appears short, dynamic cellular processes occur at the time scale of seconds and are ongoing during the labelling reaction. It is conceivable, therefore, that within this 1 min time frame, ARHGAP29 cycles on and off the rear membrane (kiss and run). This might allow ARHGAP29 to be biotinylated by Cav1-APEX2, resulting in its identification by MS. Interestingly, ARHGAP29 contains an N-terminal F-BAR domain that is required for membrane targeting and that can sense changes in membrane curvature. Work in *Caenorhabditis elegans* suggests that membrane recruitment of SPV-1, the worm orthologue of ARHGAP29, is favoured at low membrane tension, leading to RhoA inactivation at the cortex. Increased membrane tension releases ARHGAP29 from the membrane, permitting cortical RhoA activation and contractility (*Tan and Zaidel-Bar, 2015*). This is line with our proteomics data and suggests that subtle changes in membrane tension might control ARHGAP29 levels at the cell rear. Recruitment of ARHGAP29 to the rear at low membrane tension (which presumably occurs immediately following the retraction of the rear) would transiently suppress Rho-mediated contractility, providing a potential mechanism to terminate rear retraction and to permit a new protrusion/retraction cycle.

Although our data do suggest a reciprocal functional connection between Cav1 and ARHGAP29, how and where ARHGAP29 functions in RPE1 cells remains unclear. Hence, the model proposed above is largely speculative and alternative scenarios are conceivable. Firstly, the proposed model suffers from the lack of strong evidence that ARHGAP29 is actually recruited to and active at the rear. Instead of transiently operating at the rear where RhoA-GTP levels are relatively high, it is equally possible that ARHGAP29 inhibits Rho activation at the cell front, similar to what has been suggested for podocytes (*Rogg et al., 2023*). In this scenario, ARHGAP29 might be involved in establishing a Rac/Rho activity gradient along the front/rear axis (*Ridley et al., 2003*) rather than dampening RhoA activity at the rear. Secondly, we find that siRNA-mediated silencing of Cav1 expression results in a drastic reduction in cellular pMLC levels. By contrast, and unexpectedly, downregulation of ARHGAP29 had little or no effect on pMLC levels. This suggests that Cav1 and ARHGAP29 control actin dynamics in fundamentally different ways. Whilst Cav1 promotes cell rear retraction via RhoA/ROCK/MLC signalling (*Hetmanski et al., 2019*; *Shi et al., 2021*), ARHGAP29 may control the stability or turnover of F-actin networks by suppressing a RhoA/LIMK/cofilin pathway (*Qiao et al., 2017*). Cofilin-driven actin depolymerisation plays an important role in cell motility (*Bravo-Cordero et al., 2013*), although there is conflicting data over whether cofilin is active at the leading edge (*Bisaria et al., 2020*) or at the cell rear (*Mseka and Cramer, 2011*). Since myosin-II and cofilin crosstalk to control actomyosin dynamics (*Wiggan et al., 2012*), the functions of ARHGAP29 and caveolae may be intertwined in a complex manner. Thirdly, we find that downregulation of ARHGAP29 significantly increases Cav1 Y14 phosphorylation, whilst downregulation of Cav1 stabilises or promotes the expression of ARHGAP29. Previous studies have shown that Y14 phosphorylation is dependent upon ROCK and Src activity and a role in the control of focal adhesion dynamics, cell migration, and caveolae-mediated mechanosignalling has been proposed (*Joshi et al., 2012*; *Joshi et al., 2008*; *Meng et al., 2017*; *Shi et al., 2021*; *Zimnicka et al., 2016*). Taken together this suggests that ARHGAP29 interferes with RhoA/ROCK or Src signalling to suppress Cav1 phosphorylation. Caveolae, in turn, somehow dampen ARHGAP29 expression to reinforce a positive feedback loop that promotes actomyosin-mediated cell rear retraction.

In conclusion, our data and previous work (*Hetmanski et al., 2019*; *Hetmanski et al., 2021*) suggest that caveolae act as a rear-localised membrane tension sensor that links the F-actin cortex and actomyosin contractility to cell rear retraction. The caveolar interactome identified here will be instrumental in dissecting the molecular machinery and mechanisms underlying caveolae-mediated mechanotransduction in migrating cells.

## Materials and methods

### Key resources table

| Reagent type (species) or resource | Designation | Source or reference | Identifiers | Additional information |
|---|---|---|---|---|
| Cell line (human) | hTERT RPE-1 | ATCC | ATCC CRL-4000 RRID:CVCL_4388 | Human retinal pigment epithelium |
| Cell line (human) | hTERT RPE-1 Cavin1-APEX2-EGFP | *Ludwig, 2020* | NA | Stable transfection |
| Cell line (human) | hTERT RPE-1 Cavin3-miniSOG-mCherry | *Ludwig et al., 2013* | NA | Stable transfection |
| Cell line (human) | hTERT RPE-1 NES-A2E | This study | NA | Stable transfection |
| Cell line (human) | hTERT RPE-1 Cav1-A2E | This study | NA | Stable transfection |
| Transfected construct (human) | NES-A2E | *Tan et al., 2020b* | NA | Recombinant DNA |
| Transfected construct (human) | Cav1-A2E | *Ludwig et al., 2016* | NA | Recombinant DNA |
| Chemical compound, drug | DMEM:F12 1:1 Mixture | Westburg | Cat# LO BE04-687F/U1 | NA |
| Chemical compound, drug | Geneticin Selective Antibiotic (G418 Sulfate) | Fisher Scientific | Cat# 10131027 | 500 µM |
| Peptide, recombinant protein | Gibco Fibronectin Bovine Protein, Plasma | Fisher Scientific | Cat# 33010018 | 1:200 |
| Other | Lipofectamine 3000 Reagent | Thermo Fisher Scientific | Cat# L3000015 | transfection reagent |
| Other | Dharmafect 1 | Thermo Fisher Scientific | Cat# T-2001 | transfection reagent |
| Peptide, recombinant protein | Streptavidin Alexa Fluor 568 | Thermo Fisher Scientific | Cat# S11226 RRID:AB_2315774 | IF: 1:1000 |
| Peptide, recombinant protein | Streptavidin-HRP | Thermo Fisher Scientific | Cat# 434323 RRID:AB_2619743 | WB: 1:10,000 |
| Commercial assay or kit | Duolink In Situ Orange Starter Kit Mouse/Rabbit | Sigma-Aldrich | Cat# DUO92102-1KT | NA |
| Chemical compound, drug | Biotin phenol | Iris Biotech | Cat# LS-3500 | 0.5 mM |
| Chemical compound, drug | Trolox | Sigma-Aldrich | Cat# 238813 | 5 mM |
| Chemical compound, drug | Glutaraldehyde (32%) | Electron Microscopy Sciences (EMS) | Cat# 16220 | 2% |
| Chemical compound, drug | Paraformaldehyde (32%) | Electron Microscopy Sciences (EMS) | Cat# 100504–858 | 1–4% |
| Chemical compound, drug | Diaminobenzidine (DAB) (Free-Base) | Sigma-Aldrich | Cat# D8001 | 0.5 mg/ml |
| Chemical compound, drug | Durcupan ACM resin | Electron Microscopy Sciences (EMS) | Cat# 44610 | NA |
| Chemical compound, drug | Osmium tetroxide (2%) | Electron Microscopy Sciences (EMS) | Cat# 19152 | 1–2% |
| Chemical compound, drug | Potassium ferricyanide | Sigma-Aldrich | Cat# 702587 | 1–2% (w/v) |
| Peptide, recombinant protein | Sequencing Grade Modified Trypsin | Promega Corporation | Cat# V5111 | |
| Peptide, recombinant protein | Lysyl Endopeptidase, MS Grade | FUJIFILM Wako Pure Chemical Corporation | Cat# 125–05061 | |

| Reagent type (species) or resource | Designation | Source or reference | Identifiers | Additional information |
|---|---|---|---|---|
| Peptide, recombinant protein | Complete, EDTA-free Protease Inhibitor Cocktail | Roche | Cat# 11873580001 | |
| Peptide, recombinant protein | Thermo Scientific Pierce Streptavidin Magnetic Beads | Fisher Scientific | Cat# 10615204 | |
| Commercial assay or kit | EZQ Protein Quantitation Kit | Fisher Scientific | Cat# R33201 | |
| Antibody | Rabbit monoclonal anti-PARG1 (ARHGAP29) | Invitrogen | Cat# PA5-55336 RRID:AB_2645210 | WB: 1:1000 |
| Antibody | Rabbit polyclonal anti-Caveolin-1 (CAV1) | BD Bioscience/ Transduction labs | Cat# 610060 RRID:AB_397472 | WB: 1:10,000 IF/PLA: 1:500 |
| Antibody | Mouse monoclonal anti-Caveolin-1 pY14 | BD Bioscience/ Transduction labs | Cat# 611339 RRID:AB_398863 | WB: 1:1000 |
| Antibody | Rabbit polyclonal anti-Cavin-1 (PTRF) | Abcam | Cat# ab48824 RRID:AB_882224 | WB: 1:5000 IF/PLA: 1:500 |
| Antibody | Goat polyclonal anti-EHD2 | Abcam | Cat# ab23935 RRID:AB_2097328 | WB: 1:2000 |
| Antibody | Mouse monoclonal anti-Filamin 1 (E-3) (FLNA) | Santa Cruz Biotechnology | Cat# sc-17749 RRID:AB_627606 | IF/PLA: 1:100 |
| Antibody | Mouse monoclonal anti-Cortactin (clone 4F11) | Merck Millipore | Cat# 05–180 RRID:AB_309647 | IF/PLA: 1:100 |
| Antibody | Mouse monoclonal anti-Rock-1 (G-6) | Santa Cruz Biotechnology | Cat# sc-17794 RRID:AB_628223 | IF/PLA: 1:100 |
| Antibody | Mouse monoclonal anti-HSP27 (F-4) (HSPB1) | Santa Cruz Biotechnology | Cat# sc-13132 RRID:AB_627755 | IF/PLA: 1:100 |
| Antibody | Mouse monoclonal anti-GFP | Roche | Cat# 11814460001 RRID:AB_390913 | WB: 1:2000 |
| Antibody | Rabbit polyclonal anti-GFP | Abcam | Cat# ab290 RRID:AB_2313768 | IF/PLA: 1:100 |
| Antibody | Mouse monoclonal anti-YAP | Cell Signalling Technology | Cat# 14074 RRID:AB_2650491 | WB: 1:1000 |
| Antibody | Mouse monoclonal anti-YAP pS127 | Cell Signalling Technology | Cat# 13008 RRID:AB_2650553 | WB: 1:5000 |
| Antibody | Rabbit polyclonal anti-ppMLC2 (Thr18/Ser19) | Cell Signalling Technology | Cat# 3674 RRID:AB_2147464 | WB: 1:1000 |
| Antibody | Mouse anti-GAPDH | Santa Cruz Biotechnology | Cat# 47724 RRID:AB_627678 | WB: 1:5000 |
| Antibody | Donkey anti-Mouse IgG (H+L) Alexa Fluor 488 | Invitrogen | Cat# A-21202 RRID:AB_141607 | IF: 1:500 |
| Antibody | Donkey anti-Rabbit IgG (H+L) Alexa Fluor 488 | Invitrogen | Cat# A-21206 RRID:AB_2535792 | IF: 1:500 |
| Antibody | Donkey anti-Mouse IgG (H+L) Alexa Fluor 555 | Invitrogen | Cat# A-31570 RRID:AB_2536180 | IF: 1:500 |
| Antibody | Donkey anti-Rabbit IgG (H+L) Alexa Fluor 555 | Invitrogen | Cat# A-31572 RRID:AB_162543 | IF: 1:500 |
| Antibody | Goat anti-Rabbit IgG (H+L) Alexa Fluor 633 | Invitrogen | Cat# A-21071 RRID:AB_2535732 | IF: 1:500 |
| Antibody | Goat anti-Mouse IgG (H+L) Alexa Fluor 633 | Invitrogen | Cat# A-21052 RRID:AB_2535719 | IF: 1:500 |

| Reagent type (species) or resource | Designation | Source or reference | Identifiers | Additional information |
|---|---|---|---|---|
| Antibody | Goat anti-rabbit IgG (H+L) HRP conjugate | Invitrogen | Cat# A16104 RRID:AB_2534776 | WB: 1:5000 |
| Antibody | Goat anti-mouse IgG (H+L) HRP conjugate | Invitrogen | Cat# A16072 RRID:AB_2534745 | WB: 1:5000 |
| Recombinant DNA reagent | Caveolin-1 siRNA ON-TARGET Plus SMART pool | Dharmacon | Cat# L-003467-00-0020 | 100 nM |
| Recombinant DNA reagent | ARHGAP29 Mission esiRNA | Sigma | Cat# EHU13457 | 100 nM |
| Recombinant DNA reagent | FLUC Control Mission esiRNA | Sigma | Cat# EHUFLUC | 100 nM |
| Recombinant DNA reagent | EHD2 For (HindIII) | Integrated DNA Technologies | CGCAAAGCTTCTATGTTCA GCTGGCTGAAGCGG | Forward primer for cloning of EHD2-A2E |
| Recombinant DNA reagent | EHD2 Rev (EcoRI) | Integrated DNA Technologies | ATACGAATTCTCTCGGCG GAGCCCTTGTGGCG | Reverse primer for cloning of EHD2-A2E |
| Recombinant DNA reagent | CCN1 (NM_001554.5) | Integrated DNA Technologies | For: TGAAGCGGCTCCCTGTTTTT Rev: TGAGCACTGGGACCATGAAG | primers for qPCR |
| Recombinant DNA reagent | CCN2 (NM_001901.4) | Integrated DNA Technologies | For: CACCCGGGTTACCAATGACA Rev: GGATGCACTTTTTGCCCTTCTTA | primers for qPCR |
| Recombinant DNA reagent | ANKRD1 (NM_014391.3) | Integrated DNA Technologies | For: TAGCGCCCGAGATAAGTTGC Rev: GTCTGCCTCACAGGCGATAA | primers for qPCR |
| Recombinant DNA reagent | YAP (NM_001130145.3) | Integrated DNA Technologies | For: ACTCGGCTTCAGGTCCTCTT Rev: GGTTCATGGCAAAACGAGGG | primers for qPCR |
| Recombinant DNA reagent | ARHGAP29 (NM_001328664.2) | Integrated DNA Technologies | For: ACATCTAAAGCGGGTAGTAG Rev: AAGGGAGGAGATGGTGATAG | primers for qPCR |
| Recombinant DNA reagent | CAV1 (NM_001753.5) | Integrated DNA Technologies | For: ACGTAGACTCGGAGGGACA Rev: TCGTACACTTGCTTCTCGCT | primers for qPCR |
| Recombinant DNA reagent | GAPDH (NM_002046.7) | Integrated DNA Technologies | For: TCGGAGTCAACGGATTTGGT Rev: TGAAGGGGTCATTGATGGCA | primers for qPCR |
| Software, algorithm | ImageJ/FIJI | NA | https://Imagej.nih.gov/ij RRID:SCR_002285 | |
| Software, algorithm | MaxQuant (version 1.6.7.0) | NA | https://www.maxquant.org/ RRID:SCR_014485 | |
| Software, algorithm | UniProt | NA | https://www.uniprot.org/ RRID:SCR_002380 | |
| Software, algorithm | ProTIGY (version 1.1.7) | Broad Institute, Proteomics Platform | https://github.com/broadinstitute/protigy | |
| Software, algorithm | Cytoscape | NA | https://cytoscape.org RRID:SCR_003032 | |
| Software, algorithm | BioGRID | NA | https://thebiogrid.org RRID:SCR_007393 | |
| Software, algorithm | STRING | NA | https://string-db.org RRID:SCR_005223 | |
| Software, algorithm | Inkscape | NA | https://inkscape.org/ RRID:SCR_014479 | |
| Software, algorithm | R (version 4.0.3) | NA | https://www.r-project.org/ RRID:SCR_001905 | |

## DNA constructs and bacterial strains

The cavin3-miniSOG-mCherry, Cav1-APEX2-EGFP, and NES-APEX2-EGFP vectors were described previously (*Ludwig et al., 2013*; *Ludwig et al., 2016*; *Tan et al., 2020b*). To produce EHD2-A2E,

human EHD2 cDNA was cloned into the N1-A2E vector (*Tan et al., 2020b*) using PCR and *HindIII* and *XhoI* sites (see Key Resources Table for oligonucleotides used). To produce EGFP-APEX2-ARHGAP29, the ARHGAP29 cDNA was sub-cloned into the C1-A2E vector by PCR using HA-ARHGAP29 (Addgene plasmid #104154) as a template (*Tan et al., 2020b*). NEB10-beta competent *Escherichia coli* (NEB, Cat# C3019I) were transformed for plasmid amplification and then cultured in LB Agar. Single colonies were isolated and further expanded in LB broth before plasmid purification with miniprep (QIAGEN) following the manufacturer's protocol. All plasmids were verified by Sanger sequencing.

## Antibodies
All antibodies used are listed in the Key Resources Table.

## RPE1 cell culture and transfection
Human (h)TERT-RPE1 (CRL-4000) cells were purchased from ATCC and tested for mycoplasma on a regular basis using a PCR-based mycoplasma test kit. RPE1 cells were cultured in DMEM-F12 supplemented with 10% FBS, 100 units/ml penicillin, and 100 μg/ml streptomycin (Gibco) at 37°C and 5% $CO_2$. Transfections were performed in 6-well plates with the cells in a sub-confluent condition. Cells were transfected with Lipofectamine 3000 (Thermo Fisher Scientific) using 1 μg plasmid DNA per well. Stable cell lines were generated by selection in complete medium containing 400 μg/ml geneticin (Gibco) for at least 14 days. Stable cell lines were maintained in media containing 200 μg/ml geneticin and occasionally subjected to fluorescence-activated cell sorting (FACS) to maintain populations in which ~40% of cells expressed Cav1-A2E or NES-A2E in the respective lines.

## Hypo-osmotic shock and inhibitors
For hypo-osmotic shock, cells were shifted from normal growth medium to hypo-osmotic medium (10% DMEM/90% deionised water) for 5 min and then immediately collected for the hypo-osmotic condition, or switched back to normal medium for 30 min to let cells recover from the osmotic shock. For the ROCK inhibition assays, the 5 min shock was performed in the presence or absence of 10 μM Y27632 (DMSO (1 μl/ml) was used as a control), followed by 30 min recovery in the presence or absence of the inhibitor.

## Immunofluorescence
RPE1 cells grown on fibronectin-coated glass coverslips were washed twice in PBS and fixed in 4% paraformaldehyde for 15 min. After further washing, cells were permeabilised with 0.1% Triton X-100 in PBS for 5 min. Blocking was performed with 10% FBS in PBS for 1 hr at RT. Samples were incubated for 1 hr at RT with primary antibodies diluted in 0.1% bovine serum albumin (BSA), 0.01% Tween-20 in PBS (antibody buffer). After three washes with antibody buffer, samples were further incubated with the appropriate fluorescent-conjugated secondary antibody for 1 hr at RT. After three last washes in PBS, coverslips were mounted in Vectashield Antifade Mounting Medium containing DAPI (Vector Laboratories). Slides were imaged either on a spinning disk microscope (CorrSight, Thermo Fisher Scientific) equipped with an Orca R2 CCD camera (Hamamatsu) or on a laser scanning confocal microscope (LSM880 FastAiry, Carl Zeiss). Images were acquired with a ×63 oil immersion Plan-Apochromat objective (NA = 1.4, Zeiss) and standard filter sets. Confocal stacks were processed in ImageJ/FIJI.

## Electron microscopy
For electron microscopy, cells were fixed in 2.5% glutaraldehyde (EM grade, EMS) in 0.1 M cacodylate buffer pH 7.4 (CB) for 1 hr. After several washes in CB, cells were incubated in 0.5 mg/ml diaminobenzidine free-base (DAB, Sigma-Aldrich) and 0.5 mM $H_2O_2$ in CB for 5–10 min, post-fixed in 1% osmium tetroxide (EMS), 1% (v/w) potassium ferricyanide (Sigma-Aldrich) in CB, dehydrated using a graded ethanol series, and further processed for TEM as described previously (*Ludwig, 2020*). For CLEM, areas containing the cells of interest were sawed out using a jeweller's saw and sectioned parallel to the substratum using a diamond knife. 70–80 nm semithin serial sections were picked up on formvar- and carbon-coated EM slot grids and imaged on a TecnaiT12 TEM (Thermo Fisher Scientific) operated at 120 kV equipped with a 4k × 4k Eagle camera (Thermo Fisher Scientific). MiniSOG labelling and 3D tomography were described previously (*Ludwig et al., 2013*). Tomograms were reconstructed using weighted back projection and further analysed in IMOD/etomo.

## Proximity ligation assay

In situ PLA was performed with DuoLink PLA kit (Sigma-Aldrich) following the manufacturer's protocol. Briefly, after fixation and permeabilisation steps described above, cells were first incubated in Duolink Blocking Solution for 1 hr at 37°C and then with the appropriate mix of primary antibodies for 1 hr at RT. After washes, coverslips were incubated with PLA probe solution for 1 hr at 37°C. Ligation and amplification steps were performed at 37°C for 30 and 100 min, respectively. Coverslips were mounted with Duolink PLA Mounting Medium with DAPI.

## PLA imaging and quantification

Slides were imaged on a Zeiss LSM 880 confocal microscope and images were acquired with a ×63 oil immersion Plan-Apochromat objective 1.4 numerical aperture (Zeiss) and standard filter sets. Image analysis was performed with ImageJ/FIJI according to published protocols. Briefly, single-stack images were split in separate channels. The green channel was used for cell boundaries identification while the red channel for PLA signal retrieval. Thirty different cells from three independent experiments were used for quantification. The dot plots represent the number of PLA signals per cell.

## Proximity biotinylation

The APEX2 labelling protocol was adapted from previously published papers (*Hung et al., 2016*; *Tan et al., 2020a*; *Tan et al., 2020b*). Briefly, cells were grown on glass coverslips (for IF) or on plastic dishes (for MS and WB). Cells were incubated for 30 min at 37°C in media containing 0.5 mM biotin phenol (Iris Biotech). After two washes in PBS, the labelling reaction was performed for 1 min with 0.5 mM $H_2O_2$ in PBS. Due to the higher expression level of NES-A2E compared to Cav1-A2E, we adjusted the hydrogen peroxide concentration to obtain an overall biotinylation intensity comparable to that obtained in the Cav1-A2E cell line. To quench the reaction, three washes with a quencher solution (5 mM (±)–6-hydroxy-2,5,7,8-tetramethylchromane-2-carboxylic acid (Trolox), 10 mM sodium ascorbate, 10 mM sodium azide in PBS) were performed. Cells were then further processed either for IF or protein extraction.

## Protein extraction and western blotting

Cells were incubated in lysis buffer (1% SDC in 50 mM ammonium bicarbonate [AmBic]) and scraped using a cell scraper. Lysates were then sonicated (three cycles of pulse sonication for 20 s) and clarified (centrifugation at 16,000 × $g$ for 30 min at 4°C). Protein concentration was assessed either with Bradford's assay or with EZQ protein quantification assay (Thermo Fisher Scientific) and then kept at –80°C until further processing. Proteins were separated on precast 4–12% polyacrylamide gels and blotted on polyvinylidene difluoride (PVDF) membranes. Membranes were then dehydrated in methanol for 10 s and allowed to dry for 1 hr. Membranes were incubated with primary antibodies diluted in 2% BSA in PBST (PBS, 0.2% Tween-20). After 3 × 10 min washes in PBST, membranes were incubated with HRP-conjugated secondary antibody diluted in 5% fat-free milk in PBST. After further washes, membranes were developed with chemiluminescent substrate. To test biotinylation efficiency, membranes were incubated with streptavidin-HRP conjugated diluted 1:10,000 in PBST containing 2% BSA for 1 hr at RT and then washed five times with PBST before development.

## Streptavidin purification and on-bead digestion

Streptavidin Beads (Pierce Streptavidin magnetic beads, Thermo Fisher Scientific) were washed twice with cold lysis buffer before incubation with protein lysates (50 µl bead slurry per 1 mg lysate) for 1.5 hr at 4°C on a rotating wheel. Beads were then washed twice with cold lysis buffer, and then once with each of the following washing buffers: 1 M KCl, 0.1 M of $Na_2CO_3$, 2 M urea in 50 mM AmBic pH 8. The last two washes were performed with cold 50 mM AmBic pH 8. Beads were then re-suspended in digestion buffer (2 M urea in 50 mM AmBic). Protein reduction was performed by adding 1 M DTT to reach 10 mM final concentration, and incubating for 45 min at 37°C and 800 rpm. After incubation with Iodocaetamide (25 mM final concentration, 30 min at RT and 800 rpm), alkylation reaction was stopped by incubating the samples at natural light for 2 min. Protein digestion was performed in a two-step reaction: samples were pre-digested with 0.5 µg LysC for 4 hr at 37°C and 800 rpm and then digested with 1 µg trypsin O/N at 37°C and 800 rpm. Peptides were then acidified (pH 2) by adding formic acid to 1% final concentration and centrifuged for 15 min at 16,000 rpm at 4°C to remove any

residues of beads before proceeding with desalting on Sep-Pak µ-C18 Elution Plates. Peptides were then dried on a speed vac for 2 hr and re-suspended in 0.1% formic acid prior to LC-MS.

## Mass spectrometry

LC-MS/MS was performed using an Ultimate 3000 RSLCnano (Thermo Fisher Scientific) LC system equipped with an Acclaim PepMap RSLC column (15 cm × 75 µm, C18, 2 µm, 100 Å, Thermo Fisher Scientific). Peptides were eluted at a flow rate of 300 nl/min in a linear gradient of 2–35% solvent B over 70 min (solvent A: 0.1% formic acid; B: 0.1% formic acid in acetonitrile [ACN]), followed by a 5 min washing step (90% solvent B) and a 10 min equilibration step (2% solvent B). Mass spectrometry was performed on a Q-Exactive Plus mass spectrometer (Thermo Fisher Scientific) equipped with a nano-electrospray source and using uncoated SilicaTips (12 cm, 360 µm o.d., 20 µm i.d., 10 µm tip i.d.) for ionisation, applying a 1500 V liquid junction voltage at 250°C capillary temperature. MS/MS analysis were performed in data-dependent acquisition mode, and the 12 most intense parent ions were fragmented with a resolving power of 17,500 (at 200 m/z). Automatic gain control, maximum fill time, and dynamic exclusion were set to 1e6, 60 ms, and 30 ms, respectively.

## Data analysis for mass spectrometry

Protein quantification was performed with MaxQuant (version 1.6.7.0) using the following parameters: carbamidomethyl cysteine as fixed modification, methionine oxidation and N-acetylation as variable modifications, digestion mode trypsin specific with a maximum of two missed cleavages, and an initial mass tolerance of 4.5 ppm for precursor ions and 0.5 Da for fragment ions. All experiments were performed in triplicates, and the abundance of assembled proteins was determined using label-free quantification (standard MaxQuant parameters) and match between run option was selected. Protein identification was performed searching against the UniProt reference proteome (UniProt ID 9606, downloaded on January 6, 2020). To this database were added two sequences for the detection of the fusion protein: Cav1-CANLF (UniProt id: P33724) and a manually curated fusion protein combining APEX2 (Q43758) and EGFP (P42212). Further processing with Perseus (version 1.6.10.45) was performed according to the following pipeline: removal of common contaminants, reverse and peptides identified only on site-specific modifications, and log2 transformation. Only proteins identified in three out of three replicates of at least one sample were considered for further analysis. Moderated *t*-test was performed with Proteomics Toolset for Integrative Data Analysis (ProTIGY, Broad Institute, Proteomics Platform, https://github.com/broadinstitute/protigy), and a p-value <0.05 was considered statistically significant. Further data analysis and visualisation were performed in R (version 4.0.3). The mass spectrometry data is summarised in *Supplementary file 1*.

## Generation of the Cav1 interactome

Interactions among the proteins identified in the Cav1 proximity proteome were retrieved from BioGRID version 4.2.193 (April 2020) and the STRING database. For BioGRID entries, only physical protein interactions were considered; methods such as Affinity Capture-RNA, Protein-RNA, and Proximity Label-MS were disregarded. For STRING, the following parameters were used: Active interaction; source: Database; Confidence: >0.4. In addition, direct or indirect protein–protein interactions reported in the primary literature were included (see *Supplementary file 2*). The protein network was drawn in Cytoscape version 3.7.1.

## Live-cell imaging and cell migration analysis

RPE1 cells were grown on fibronectin-coated glass-bottom dishes (MatTec Corp., Ashland, MA) and imaged live in complete DMEM without Phenol Red on a CorrSight spinning disk microscope (Thermo Fisher Scientific). Imaging was carried out at 37°C, 5% $CO_2$, and 90% humidity in a closed atmosphere chamber (IBIDI). Up to five different locations were imaged simultaneously using the multi-stage position function in LA software (Thermo Fisher Scientific). Confocal image stacks (typically 3–4 images with a z-interval of 600–1000 nm) were acquired as 3 × 3 tiles with a ×40 oil objective (NA 1.3, EC Plan Neofluar M27, Zeiss) using the 488 nm laser line (65 mW; iChrome MLE-LFA) and a standard GFP filter set. Typically time-lapse movies were recorded at a frame rate of 5 or 10 min for 6–12 hr (in some cases even longer). Focus was maintained by a hardware autofocus system (Focus Clamp). The laser output power and exposure times were set to a minimum. Tiles were stitched and confocal image

stacks converted into maximum intensity projections. Time-lapse recordings were analysed in ImageJ/FIJI (*Schindelin et al., 2012*; *Schneider et al., 2012*).

Cell migration data was analysed with the ImageJ Manual Tracking plugin. Twenty-five cells per condition in five individual movies were analysed, and all experiments were performed in triplicate. GFP-positive cells were identified and tracked frame-by-frame by manually clicking on the centre of the nucleus. The X/Y coordinates were then imported into R, and CelltrackR package was used for further analysis (*Wortel et al., 2021*). Data were normalised to overlay track starting points to zero. The R package was then used to calculate the duration of migration, displacement, track length, speed, and mean squared displacement.

### Quantification of rear localisation

All quantifications were performed in ImageJ/FIJI. For the quantification of Cav1 rear enrichment in fixed cells shown in *Figure 1A*, the mean pixel intensities at the cell rear and the cell front were measured. The cell rear was outlined manually based on the highest Cav1 signal. An equal size area was then measured at the front of the cell. The rear/front signal intensity ratios for each cell were calculated and plotted.

For the quantification of caveolae rear localisation in time-lapse movies, a 50 pixel wide line was drawn from the rear towards the front along the longest cell axis. The position and vector of the line were adjusted for each frame. Pixel intensities along the line were measured using the Plot Profile function of ImageJ/FIJI and exported into Excel. For the quantification shown in *Figure 5D*, the pixel intensities along the line were divided into four equal size quadrants - rear, centre rear, centre front, and front. The average pixel intensities in each quadrant were determined and plotted. All pixel intensity values were background subtracted. Graphs were produced in Excel, Prism, or R.

### siRNA transfection

siRNAs specific to human caveolin-1 and ARHGAP29 and non-targeting siRNAs (see Key Resources Table) were transfected at a final concentration of 50–100 nM using Dharmafect (Sigma-Aldrich) according to the manufacturer's recommendations. Transfected cells were analysed by western blotting 48 hs post-transfection. For immunofluorescence and cell migration analysis, cells were trypsinised 48 hr post-transfection and seeded at low density onto glass coverslips or MatTec dishes, respectively. Cells were imaged 24 hr later.

### qPCR analysis

RNA was extracted using the PureLink RNA Mini Kit (Invitrogen). Genomic DNA was removed using DNAse1 (Thermo Scientific) and cDNA was prepared using Superscript II Reverse Transcriptase (Invitrogen). qPCR was carried out using SsoAdvanced Universal SYBR Green Supermix (Bio-Rad) on a CFX96 Real-Time System (Bio-Rad). Experiments were conducted in technical triplicates. Data was normalised to endogenous GAPDH levels and analysed using the comparative Ct method. Error bars represent the mean ± standard deviation, and a Student's *t*-test was used to generate p-values (*≤0.05, **≤0.01, ***≤0.001) (qPCR primers are listed in the Key Resources Table).

## Acknowledgements

RG and EM were supported by the FNR AFR Bilateral Singapore grant 11823257 to GD and AL. AL was supported by a start-up grant from the Nanyang Technological University. We thank Barbara Hübner and Linda Jiabao for their help during the revision of this manuscript.

## Additional information

### Funding

| Funder | Grant reference number | Author |
| --- | --- | --- |
| Fonds National de la Recherche Luxembourg | 11823257 | Gunnar Dittmar Alexander Ludwig |

| Funder | Grant reference number | Author |
| --- | --- | --- |
| Nanyang Technological University | Start-up | Alexander Ludwig |

The funders had no role in study design, data collection and interpretation, or the decision to submit the work for publication.

## Author contributions

Eleanor Martin, Rossana Girardello, Data curation, Formal analysis, Validation, Investigation, Visualization, Methodology; Gunnar Dittmar, Conceptualization, Resources, Supervision, Funding acquisition, Project administration; Alexander Ludwig, Conceptualization, Resources, Data curation, Formal analysis, Supervision, Funding acquisition, Investigation, Visualization, Methodology, Writing - original draft, Project administration, Writing - review and editing

## Author ORCIDs

Eleanor Martin (iD) http://orcid.org/0000-0003-4707-697X
Rossana Girardello (iD) http://orcid.org/0000-0001-8605-6037
Gunnar Dittmar (iD) http://orcid.org/0000-0003-3647-8623
Alexander Ludwig (iD) https://orcid.org/0000-0002-0696-5298

## Decision letter and Author response

Decision letter https://doi.org/10.7554/eLife.85601.sa1
Author response https://doi.org/10.7554/eLife.85601.sa2

---

# Additional files

## Supplementary files

- Supplementary file 1. Mass spectrometry data.
- Supplementary file 2. Literature and database entries for the Cav1 interactome analysis.
- MDAR checklist

## Data availability

The raw mass spectrometry data was deposited in the PRIDE repository under accession PXD026464.

The following dataset was generated:

| Author(s) | Year | Dataset title | Dataset URL | Database and Identifier |
| --- | --- | --- | --- | --- |
| Martin E, Girardello R, Dittmar G, Ludwig A | 2024 | Spatially restricted proteomics reveals dynamic changes in the caveolin-1 interactome in response to increased membrane tension | https://www.ebi.ac.uk/pride/archive/projects/PXD026464 | PRIDE, PXD026464 |

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
