## [Editor Report]

This important study uses powerful time-resolved proximity proteomics, validated with proximity ligation assays, to provide new insights into the mechanical regulation of caveolin-1 complexes that form in migrating cells. Follow-up experiments reveal a reciprocal relationship between mechanosensitive caveolae and RhoGTPase signaling in migrating cells. This work is generally convincing, with the exception that the functional link between ARHGAP29 and caveolae under low membrane tension would benefit from a more detailed characterization in the future

---

## [Decision Letter]

**Decision letter after peer review:**

Thank you for submitting your article "Time-resolved proximity proteomics uncovers a membrane tension-sensitive caveolin-1 interactome at the rear of migrating cells" for consideration by *eLife*. Your article has been reviewed by 2 peer reviewers, and the evaluation has been overseen by a Reviewing Editor and Jonathan Cooper as the Senior Editor. The following individual involved in review of your submission has agreed to reveal their identity: Patrick T Caswell (Reviewer #1).

Essential revisions (for the authors):

You will see in the comments below that the Reviewers are enthusiastic about your proteomics approach, but have a few concerns about the follow-up experiments. We invite you to submit a revised manuscript in light of all their comments, but with particular consideration of the following points:

1) We would like you to better quantify caveolae formation at the cell scale, as there is some potential discrepancy between some Figures. Furthermore, you frequently present the osmotic shock as a "high membrane tension" stimulus, but the study focuses on the rear of the cell and it isn't clear whether a global mechanical stimulus triggered by an osmotic shock really informs on what is happening locally. We all agree that a better characterization of the "front-to-back" cell model is needed.

2) We would also like to see a better characterization of the link between ARHGAP29 and caveolae, which is not sufficiently clear at this stage.

*Reviewer #1 (Recommendations for the authors):*

CAV1-A2E levels seem consistently lower in the hypo-osmotic conditions, which could reflect a change in oligomer formation or a difference in the labelling efficiency under this condition. Is the labelling efficiency across the three conditions (NT, HYPO, REC) the same, and similar between both Cav1-APEX2-EGFP and NSE-A2E?

Figure 3E is a little misleading in my opinion, and would be improved by an indication of where significantly altered protein changes are. For example, UTRN and TNS1 appear high (by z-score) in NT versus Hypo, but are actually only found in one repeat for the NT sample (Figure 3 Supplement 1) so are unlikely to be significant.

*Reviewer #2 (Recommendations for the authors):*

1. Based on cavin3 imaging, the author found that caveolae are stably associated at the rear of the cells. While an enrichment of cavin3 at the rear is visible in Figure 1A-B, there is no analysis of caveolae localization at the whole cell level. Because caveolae are well known to come with different sizes and lifetimes, a robust quantitative of analysis of the distribution and dynamics of Cav1, cavin1, 3 by fluorescence microscopy (endogenous and overexpressed proteins) would better support this claim. In addition, an analysis of these structures between the different area of the cell (front- middle-rear) during the different state of RPE1 cells (immobile and mobile) would be helpful to better establish this model. While the front cell in Figure1 shows almost no caveolae, the phenotype looks much less dramatic in Figure2 and Figure 4. Caveolae are distributed everywhere with some enrichment at the rear.

2. The authors identified new sets of protein associated with caveolae exposed at different membrane tension using a well-designed approach based on proximity biotinylation (Cav1-APEX), a methodology used previously by the group (reference 43-44). I congratulate the authors for these compelling set of data. Though, more explanation of the specific detection of biotinylated caveolae at the rear of the cell would be helpful. Once would expect that this strategy is global and all caveolin-1 positive structures should be detected. As recommended in point 1, a careful analysis of these cells would help to determine the degree of specificity of the proteomic analysis for caveolae localized at the rear of the cells.

Secondly, it should be discussed that the level of tension induced by the hypotonic shock is likely different from the one exerted at the rear of the cell. Thus, proteomic results might vary somehow from the physiology of a migrating cell.

Moreover, it is expected that hypo conditions would increase membrane tension and promote the dissociation of caveolae as observed by others (Sinha et al. 2011). This is actually observed in Figure 3 but this is not what is observed by the authors in Figure 1 where it is stated that caveolae were stably associated at the rear of migrating cell. Could the authors comment on this difference?

Last, providing evidence of actual membrane tension at front and rear would be interesting to support the statements of this study. The link is not clear to me. While the authors state that membrane tension is higher at the rear of the cells, other evidence show that membrane tension is higher at the front cell and lower at the rear. Hetmanski et al. (2019) measured membrane tension at front and rear in migrating cells by optical tweezer and did not report significant differences, except during durotactic cells where they found lower tension membrane at the rear, which is the opposite. Authors should clarify this point in the manuscript to avoid confusion.

3. Authors use PLA experiments to validate their proteomics analysis and results are accordingly quantified by the number of PLA spots detected (interaction). I was wondering what is the level of interaction (PLA) versus the total number of total caveolin-1 spots detected in all conditions? In other words, what is the proportion of proteins interacting with the caveolae population (and where in the cell)? This analysis could help to clarify the point of the authors relative to the existence of confined subpopulations of caveolae (detected by PLA) at the rear of the cell.

4. Better characterization of RhoGA ARHGAP29 is encouraged. Complementary experiments using Knock-out, dominant negative and dominant positive mutants would help as no phenotype is observed using KD experiment. Do overexpression of RhoGA ARHGAP29 displace the interaction of other regulators of the pathway identified at caveolae? Since overexpression is transient, phenotype may also vary from cell to cell. Is there a correlation between the level of overexpression and the phenotype observed?

---

## [Author Response]

Essential revisions (for the authors):You will see in the comments below that the Reviewers are enthusiastic about your proteomics approach, but have a few concerns about the follow-up experiments. We invite you to submit a revised manuscript in light of all their comments, but with particular consideration of the following points:1) We would like you to better quantify caveolae formation at the cell scale, as there is some potential discrepancy between some Figures. Furthermore, you frequently present the osmotic shock as a "high membrane tension" stimulus, but the study focuses on the rear of the cell and it isn't clear whether a global mechanical stimulus triggered by an osmotic shock really informs on what is happening locally. We all agree that a better characterization of the "front-to-back" cell model is needed.2) We would also like to see a better characterization of the link between ARHGAP29 and caveolae, which is not sufficiently clear at this stage.

We thank the reviewers and the editor for their insightful comments and the critical evaluation of our manuscript. In the revised manuscript we have addressed point 1 comprehensively by quantifying the rear localisation of endogenous and tagged Cav1 and cavins in fixed and migrating RPE1 cells. We have also analysed and now describe in more depth the dynamic behaviour of caveolae in RPE1 cells. In addition, we have analysed the PLA data in more detail and improved the data representation in our figures. We hope this addresses several questions and concerns raised by both reviewers, in particular regarding the enrichment of caveolae at the rear.

We have also much extended our discussion: We discuss technical aspects and the interpretation of our proteomics data, we elaborate on published literature describing how membrane tension is organised and generated in migrating cells, and we provide an objective discussion on how caveolae and ARHGAP29 might function at the cell rear.

With regards to point 2, we performed several new experiments to better characterise the functional link between caveolae and ARHGAP29 in RPE1 cells. As you will see in the specific responses to the reviewer’s questions, these experiments unfortunately have been largely unsuccessful or difficult to interpret.

Please find below a detailed point-by-point response to the reviewer’s comments. The comments are in normal font, our response is in bold/yellow. Additions and changes to the manuscript are highlighted in yellow in the revised manuscript file.

Reviewer #1 (Recommendations for the authors):CAV1-A2E levels seem consistently lower in the hypo-osmotic conditions, which could reflect a change in oligomer formation or a difference in the labelling efficiency under this condition. Is the labelling efficiency across the three conditions (NT, HYPO, REC) the same, and similar between both Cav1-APEX2-EGFP and NSE-A2E?

This might be due to a misunderstanding of the data. The log2 fold change for Cav1-A2E is not noticeably different between the three samples. It’s ~4 in the NT and REC samples, and ~3 in the Hypo sample (see Figure 4). We do not interpret this as a biologically meaningful difference. What is different is the log10 p-value, which for Cav1-A2E and cavin1/PTRF is lower in the NES/Hypo sample compared to the other two samples. A lower p value however simply means there was more variance in the data measurements, for unknown reasons. What is important to note is that Cav1, Cav2 and EHD2 are no longer significantly enriched in the NES/Hypo sample. This is consistent with our conclusion that the caveolar coat disassembles under these conditions. The fact that cavin1/PTRF is still enriched in the NES/Hypo dataset may indicate that the coat is only partially disassembled, or that much of the cavin1 pool remains attached to Cav1-enriched membranes when cells are exposed to hypoosmotic shock.

Figure 3E is a little misleading in my opinion, and would be improved by an indication of where significantly altered protein changes are. For example, UTRN and TNS1 appear high (by z-score) in NT versus Hypo, but are actually only found in one repeat for the NT sample (Figure 3 Supplement 1) so are unlikely to be significant.

This is due to the way the proteomics data was analysed. We filtered the data in such a way that at least one measurement in each sample was needed for the identified protein to be included in the dataset. Some proteins, such as Pacsin2, were not measured at all in any of the three Hypo replicates. This likely reflects a strong dissociation of Pacsin2 from Cav1 under these conditions. Conversely, UTRN and TNS1 were not identified in two out of the three NT replicates, but were identified in all three REC replicates. These proteins were included because two out of the three states could be compared (NT vs REC in this case). Based on the MS data it is tempting to speculate that UTRN and TNS1 are transient components of the Cav1 interactome that are specifically needed during early stages of caveolae formation. However, since we did not have the tools (antibodies) to confirm the spatial association between UTRN, TNS1 and Cav1 using PLA, we did not include this speculation in the manuscript.

Reviewer #2 (Recommendations for the authors):1. Based on cavin3 imaging, the author found that caveolae are stably associated at the rear of the cells. While an enrichment of cavin3 at the rear is visible in Figure 1A-B, there is no analysis of caveolae localization at the whole cell level. Because caveolae are well known to come with different sizes and lifetimes, a robust quantitative of analysis of the distribution and dynamics of Cav1, cavin1, 3 by fluorescence microscopy (endogenous and overexpressed proteins) would better support this claim.

As mentioned above, we have carefully analysed the localisation of caveolae in fixed cells (using Cav1 and cavin1 antibodies as well as Cav1 and cavin fusion proteins) and in live cells transfected with Cav1 and cavin fusion proteins. The analysis clearly demonstrates an enrichment of caveolae at the rear, independent of the fusion protein used (Figure 1 and Figure 1 —figure supplement 1). Our tomography and TEM data supports the abundance of caveolae at the rear as well (Figure 2).

In addition, an analysis of these structures between the different area of the cell (front- middle-rear) during the different state of RPE1 cells (immobile and mobile) would be helpful to better establish this model. While the front cell in Figure1 shows almost no caveolae, the phenotype looks much less dramatic in Figure2 and Figure 4. Caveolae are distributed everywhere with some enrichment at the rear.

This cell-to-cell variability in caveolae distribution can be explained by the dynamic behaviour of RPE1 cells as they migrate. RPE1 cells migrate randomly and frequently reorient their front-rear axis as they turn or change direction. When this happens caveolae are redistributed, and when such cells are fixed in this particular moment in time, caveolae appear less polarised and more evenly distributed. We now describe these dynamic behaviours in the first paragraph of the Results section. We have included new figure panels (Figure 1 Figure Supplement 2) and we provide several movies that illustrate this dynamic behaviour.

2. The authors identified new sets of protein associated with caveolae exposed at different membrane tension using a well-designed approach based on proximity biotinylation (Cav1-APEX), a methodology used previously by the group (reference 43-44). I congratulate the authors for these compelling set of data. Though, more explanation of the specific detection of biotinylated caveolae at the rear of the cell would be helpful. Once would expect that this strategy is global and all caveolin-1 positive structures should be detected. As recommended in point 1, a careful analysis of these cells would help to determine the degree of specificity of the proteomic analysis for caveolae localized at the rear of the cells.

As outlined above, we have now quantified the localisation of Cav1 and cavins in fixed and live cells. We show that Cav1 is enriched ~8 fold at the rear vs the front. A similar enrichment was observed for cavin1 (see Figure 1A-1D and Figure 1 —figure supplement 1). Given this strong enrichment at the trailing edge we propose that most of the proteins we detect in the MS dataset are associated with caveolae at the rear, because this is where caveolae are most abundant. However, strictly speaking, the proteome is indeed not a specific proteome of Cav1 at the cell rear (this is now clearly stated in the first paragraph of the discussion). There is no straightforward way to deconvolve the proteome to discriminate between proteins that are specifically associated with caveolae at the rear vs other places in the cell. We have however partially addressed this question with our new PLA analysis (Figure 5G), which shows that the majority of PLA signal between the caveolar core proteins (Cav1, cavin1, EHD2) is produced at the rear (70-80%). By contrast, proteins that are present but not restricted to the rear, such as CTTN and FLNA, do not show a clear rear bias in their interactions with Cav1.

Secondly, it should be discussed that the level of tension induced by the hypotonic shock is likely different from the one exerted at the rear of the cell. Thus, proteomic results might vary somehow from the physiology of a migrating cell.

We agree that the hypoosmotic shock approach is expected to cause membrane tension changes very different to those a cell experiences during migration. This clearly is a limitation of the assay. However, the assay is widely used to address how membrane tension affects plasma membrane dynamics. It is rapid and reversible, and compatible with biochemical analysis (which was critical in our case). This assay coupled with APEX2 proteomics approach allowed us to obtain snapshots of the Cav1 interactome under resting conditions and upon an acute increase in membrane tension. Future experiments should now address how membrane tension changes are generated at the cell rear, how much tension is generated, and how caveolae respond to this. This is now discussed in the second paragraph of the discussion.

Measuring membrane tension changes in live migrating cells over time is not trivial and has not been done to our knowledge. Interestingly, however, a recent preprint by the Roux lab addresses this question using the Flipper-TR probe and FLIM. The data show that membrane tension gradients (low at the rear and high at the front) exist in migrating and even in non-migrating cells and that F-actin assembly and dynamics are critical for generating such gradients ^5^.

Moreover, it is expected that hypo conditions would increase membrane tension and promote the dissociation of caveolae as observed by others (Sinha et al. 2011). This is actually observed in Figure 3 but this is not what is observed by the authors in Figure 1 where it is stated that caveolae were stably associated at the rear of migrating cell. Could the authors comment on this difference?

Using live cell imaging we indeed find (and have now quantified in detail) that Cav1 and cavins are stably associated with the cell rear when cells migrate in 2D (Figure 1). However, given the limited resolution provided by light microscopy, it is impossible to discriminate between flat and invaginated caveolae. Electron tomography of the cell rear however confirmed that the signals we observe in light microscopy correspond to a large extent to invaginated and interconnected caveolae networks. For simplicity sake, we often refer in the text to caveolae when we describe the localisation of Cav1 or cavins. Strictly speaking however, it is likely that at any given point in time different states of caveolae co-exist at the rear. When tension is very low (for example in response to a sudden rear retraction), the formation of large invaginated caveolae networks of the kind we observe by tomography is favoured. When tension builds up (for example when front protrusion is dominant over rear retraction), a fraction of such networks or single caveolae will flatten out. These assumptions are based on data showing that protrusion at the front is coupled to retraction of the rear, and that tension can propagate within the cell ^7-9^.

Our live imaging data does indeed suggest that the levels of Cav1 and cavins fluctuate at the rear; stronger signals tended to be observed right after cell rear retraction. Conversely, when cell protrusion was dominant (i.e. when cells stretched), the rear intensity appeared to drop, which could be interpreted as caveolae disassembly. We attempted to measure the rear signal intensity over time but found that these variations were difficult to quantify. Overall, we conclude that Cav1 and cavin1 remain rear localised even when cells protrude.

The model that cavins and other caveolar proteins completely dissociate from Cav1 at high membrane tension has not been formerly proven. In fact, a recent study by Matthaeus et al. showed, using correlative platinum replica EM, that a substantial fraction of cavins and EHD2 are associated with flat Cav1 scaffolds ^10^. It is possible therefore that membrane tension-induced conformational changes in the caveolar coat (e.g. induced by hypo-osomotic shock) are more subtle than the original literature on this suggests ^11^. Nonetheless, our proteomics data and PLA assays clearly show that membrane tension elicits profound and quantifiable changes in the caveolar coat. We propose that the coat is either partially dissolved, or that a considerable fraction of caveolae undergo major remodelling. This is now discussed in the second paragraph of the discussion.

Last, providing evidence of actual membrane tension at front and rear would be interesting to support the statements of this study. The link is not clear to me. While the authors state that membrane tension is higher at the rear of the cells, other evidence show that membrane tension is higher at the front cell and lower at the rear. Hetmanski et al. (2019) measured membrane tension at front and rear in migrating cells by optical tweezer and did not report significant differences, except during durotactic cells where they found lower tension membrane at the rear, which is the opposite. Authors should clarify this point in the manuscript to avoid confusion.

We apologise for the misleading description of membrane tension gradients in the original manuscript. We intended to say that membrane tension has been reported to be lower at the rear and higher at the front. This is now better explained in the revised version. It is also true that Hetmanski et al. did not observe a difference in membrane tension between the front and the rear in cells grown on glass (uniform stiff substrates), but a clear tension differential was observed when cells were embedded in stiffness gradients, with the rear membrane showing significantly lower membrane tension than the front ^6^. Lieber et al. had previously measured membrane tension in fish keratinocytes migrating on a 2D surface, and similarly found that the rear exhibits lower membrane tension than the front ^12^. This suggests that steady-state variations in tension can exist in the plasma membranes of moving cells. However, whether such membrane tension gradients are formed in all migrating cells is still unclear. In fact, there is a lively debate over whether membrane tension can be controlled locally ^13^ or whether it equilibrates rapidly ^9^.

Our new quantifications clearly show that caveolae are enriched at the RPE1 cell rear. Such a polarized distribution is also observed in many other cell types ^14,15^. Based on this and existing literature we suggest that low membrane tension at the rear promotes caveolae formation, which was nicely demonstrated by Hetmanski et al. Interestingly, a recent preprint by the Roux lab shows that membrane tension gradients (measured using the Flipper-TR probe) exist in many migrating and even in non-migrating cells ^5^. We have included a section in the discussion in which we elaborate on this topic and on how membrane tension might regulate caveolae rear localisation and assembly. The relationship between caveolae, membrane tension, and actomyosin contractility is clearly complex and requires further work that goes beyond the objective of this manuscript.

3. Authors use PLA experiments to validate their proteomics analysis and results are accordingly quantified by the number of PLA spots detected (interaction). I was wondering what is the level of interaction (PLA) versus the total number of total caveolin-1 spots detected in all conditions? In other words, what is the proportion of proteins interacting with the caveolae population (and where in the cell)? This analysis could help to clarify the point of the authors relative to the existence of confined subpopulations of caveolae (detected by PLA) at the rear of the cell.

As described above, we have quantified the fraction of PLA dots we observe at the cell rear for four different combinations of antibodies/targets: Cav1/PTRF, Cav1/EHD2, Cav1/CTTN, and Cav1/FLNA (Figure 5G). In agreement with the localisation data, we find that 70-80% of the PLA dots for Cav1/PTRF and Cav1/EHD2 come from the cell rear, whilst this fraction is considerably lower (~40%) for Cav1/CTTN and Cav1/FLNA.

4. Better characterization of RhoGA ARHGAP29 is encouraged. Complementary experiments using Knock-out, dominant negative and dominant positive mutants would help as no phenotype is observed using KD experiment. Do overexpression of RhoGA ARHGAP29 displace the interaction of other regulators of the pathway identified at caveolae? Since overexpression is transient, phenotype may also vary from cell to cell. Is there a correlation between the level of overexpression and the phenotype observed?

We agree that the functional connection between ARHGAP29 and caveolae and the role of ARHGAP29 in migrating cells was not well defined. Although we present some evidence for a reciprocal relationship between ARHGAP29 and caveolae, we concede we have been unable to add more mechanistic insight into this relationship. We attempted to produce ARHGAP29 KO RPE1 cells but failed in several attempts. We also performed additional siRNA and overexpression experiments, which were mostly non-conclusive or difficult to interpret (see comments to Reviewer 1). With regards to the last point, we do not find a strong correlation between the level of ARHGAP29 overexpression and cell migration behaviour. That said, we note that cells expressing very high levels of ARHGAP29 often exhibited a rounded cell shape, a much reduced F-actin content, and showed no net translocation at all in time lapse imaging. Such cells were not considered in our analysis.

References:

1. Qiao Y, Chen J, Lim YB, et al. YAP Regulates Actin Dynamics through ARHGAP29 and Promotes Metastasis. Cell reports. 2017;19(8):1495-1502.

2. Rausch V, Bostrom JR, Park J, et al. The Hippo Pathway Regulates Caveolae Expression and Mediates Flow Response via Caveolae. Curr Biol. 2019;29(2):242-255 e246.

3. Hung V, Udeshi ND, Lam SS, et al. Spatially resolved proteomic mapping in living cells with the engineered peroxidase APEX2. Nat Protoc. 2016;11(3):456-475.

4. Wiggan O, Shaw AE, DeLuca JG, Bamburg JR. ADF/cofilin regulates actomyosin assembly through competitive inhibition of myosin II binding to F-actin. Dev Cell. 2012;22(3):530-543.

5. Juan Manuel García-Arcos AM, Julissa Sánchez Velázquez, Pau Guillamat, Caterina Tomba, Laura Houzet, Laura Capolupo, Giovanni D’Angelo, Adai Colom, Elizabeth Hinde, Charlotte Aumeier, Aurélien Roux. Actin dynamics sustains spatial gradients of membrane tension in adherent cells. bioRxiv 20240715603517. 2024.

6. Hetmanski JHR, de Belly H, Busnelli I, et al. Membrane Tension Orchestrates Rear Retraction in Matrix-Directed Cell Migration. Dev Cell. 2019;51(4):460-475 e410.

7. Tsai TY, Collins SR, Chan CK, et al. Efficient Front-Rear Coupling in Neutrophil Chemotaxis by Dynamic Myosin II Localization. Dev Cell. 2019;49(2):189-205 e186.

8. Mueller J, Szep G, Nemethova M, et al. Load Adaptation of Lamellipodial Actin Networks. Cell. 2017;171(1):188-200 e116.

9. De Belly H, Yan S, Borja da Rocha H, et al. Cell protrusions and contractions generate long-range membrane tension propagation. Cell. 2023.

10. Matthaeus C, Sochacki KA, Dickey AM, et al. The molecular organization of differentially curved caveolae indicates bendable structural units at the plasma membrane. Nat Commun. 2022;13(1):7234.

11. Sinha B, Koster D, Ruez R, et al. Cells respond to mechanical stress by rapid disassembly of caveolae. Cell. 2011;144(3):402-413.

12. Lieber AD, Schweitzer Y, Kozlov MM, Keren K. Front-to-rear membrane tension gradient in rapidly moving cells. Biophysical journal. 2015;108(7):1599-1603.

13. Shi Z, Graber ZT, Baumgart T, Stone HA, Cohen AE. Cell Membranes Resist Flow. Cell. 2018;175(7):1769-1779 e1713.

14. Grande-Garcia A, Echarri A, de Rooij J, et al. Caveolin-1 regulates cell polarization and directional migration through Src kinase and Rho GTPases. The Journal of cell biology. 2007;177(4):683-694.

15. Grande-Garcia A, del Pozo MA. Caveolin-1 in cell polarization and directional migration. Eur J Cell Biol. 2008;87(8-9):641-647.

16. Ludwig A, Howard G, Mendoza-Topaz C, et al. Molecular composition and ultrastructure of the caveolar coat complex. PLoS biology. 2013;11(8):e1001640.